# Quantitative use-wear analysis of stone tools: Measuring how the intensity of use affects the identification of the worked material

**Juan José Ibáñez**[1]*, **Niccolò Mazzucco**[2]

1 Archaeology of Social Dynamics (ASD), Institución Milá y Fontanals (IMF), Spanish National Research Council (CSIC), Barcelona, Spain, 2 Department of Civilizations and Forms of Knowledge, University of Pisa, Pisa, Italy

* ibanezjj@imf.csic.es

**Data Availability Statement:** All relevant data are within the manuscript and its Supporting Information files.

## Abstract

The identification of the use of stone tools through use-wear analysis was one the major methodological advances in Prehistoric Archaeology during the second half of the 20th century. Studies of use-wear analysis have decisively contributed to a better understanding of the cognitive capacities and the socio-economic organization of Prehistoric societies. Among use-wear traces, microwear polish is the most relevant evidence, as it allows the identification of the worked materials (i.e. wood, antler, hide, bone, stone...). This identification is currently carried out through the qualitative and visual comparison of experimental and archaeological tools. During the last decade, confocal microscopy is allowing the quantitative identification of the worked material through the texture analysis of microwear polish. Previous tests have accounted for the variability of use-wear traces as caused by different types of worked material. However, how the intensity of use, which is widely recognized as an important factor conditioning microwear polish characteristics, affects our capacity to identify the worked materials is poorly understood. This research addresses the dynamic nature of microwear polish through confocal microscopy and texture analysis. This research shows that use-wear polishing is a dynamic process and surface texture evolves continuously during the working time. The evolution fits a logarithmic function, so most texture modification takes place during the first phases of work. The way in which polish texture evolves through time differs from one contact material to the other. We demonstrate that, despite the dynamic nature of use-wear polish, different worked materials can be discriminated. However, some overlapping between used and unused surfaces and between worked materials occurs in the first stages of use. Moreover, polishes of similar characteristics (i.e. bone and antler) can show similar texture at advanced stages of use. These problems in identification can be in good measure overcome by creating dynamic models of polish texture in which not only the worked material but also the time of use is considered.

**Funding:** This research is part of a Marie Sklodowska Curie Individual Fellowship (Grant Number 792544) (NM). Funding was also provided by the project PID2019-105767GB-I00 (JJI) of the Spanish Ministry of Economy. We acknowledge support of the publication fee by the CSIC Open Access Publication Support Initiative through its Unit of Information Resources for Research (URICI). The funders had no role in study design, data collection and analysis, decision to publish, or preparation of the manuscript.

**Competing interests:** The authors have declared that no competing interests exist.

# Introduction

More than 60 years after the English translation of the pioneering study of the function of prehistoric tools through use-wear analysis by S. Semenov [1], his method of analysis, later developed by other scholars [2–7] is today widely applied to archaeological case studies for different periods, cultures and locations [8–12].

The method is based on the comparison of the use-wear traces generated on experimental tools and those observed on archaeological instruments. To explain it in a schematic way, micro-scarring offers information on the motion of the tool and the relative hardness of the worked material, edge rounding on the position of the tool and the abrasiveness of the worked material, striations on the tool motion, and microwear polish distribution and texture on the type of worked material (wood, antler, hide. . .). Thus, microwear polish characteristics are crucial for the identification of the tool use [2]. Traditionally, experimental and archaeological microwear polishes are matched through visual comparison.

During the 1980s, the ability to identify worked materials was fiercely disputed. Based on the poor results of a series of blind tests [13–15] and on the incapacity of their image analysis to discriminate different use-wear polishes, R. Grace and colleagues concluded that worked materials could not be identified through microwear polish analysis [16]. They pointed out that microwear polish characteristics depend mainly on the type of worked material and on the time of use (polish development). They proposed a model of use-wear polish development in which every kind of polish generated by every worked material passed through the same steps of polish characteristics, so working a soft contact material for 30 minutes would cause the same alteration as working a harder material for a shorter time. As microwear polishing during tool use is a dynamic process and the time of polish development on archaeological tools is not known, they concluded that the worked materials could not be identified. Several scholars criticized the conditions of the blind test [5, 17, 18], while the re-evaluation of the main blind tests carried out shows more positive interpretations [19, 20]. New image analysis of microwear polishes demonstrated the weaknesses of Grace's studies, and showed that use polishes from different contact materials could be discriminated [21–24]. Specialists in use-wear analysis affirmed that, when the use-wear polish was well developed, the worked material could be correctly identified [2]. Contrasting with Grace's model, Vaughan's model of polish development consists of three successive phases: generic weak polish, smooth pitted polish, and well-developed polish [3]. In this model, after the first phase of generic weak polish, each worked material develops specific use-wear polish characteristics. Thus, use-wear polish would reach a phase of stability in its development after a certain time of use [20–24]. However, Grace's doubts on the method were accepted by many archaeologists, affecting the development of the discipline especially in Anglo-Saxon countries [25].

This criticism reinforced the need to provide quantitative analytical procedures for the method of use-wear analysis. In fact, the need for quantification had long been felt by researchers; L.H. Keeley had already tried to quantify the light reflectivity of microwear polishes [2]. After him, many scholars tested different methods for quantifying microwear polish characteristics, such as interferometry [26], atomic force microscopy [27], 3D rugosimetry [28] or the above-mentioned studies based on image analysis. These studies demonstrated that microwear polish shows consistent quantitative characteristics that can be related to the worked materials. However, different technical limitations of the methods hindered their integration in the standard use-wear analytical protocol.

At the beginning of the millennium, confocal microscopy, a powerful new method for texture quantification, started to be used, first for tooth microwear analysis [29] and then for use-wear analysis of lithic tools [30]. This method successfully discriminated 3D images of

microwear polish from different contact materials in experimental tools [31–35]. Some years ago, after preliminary essays [36], it was possible to distinguish tools used for reaping four types of plants: wild cereals in natural stands, cultivated wild cereals, domestic cereals and reeds [37], to which grass cutting was later incorporated [38]. For the first time, this method was used not only to quantitatively discriminate 3D images of polish generated by different worked materials but also to identify the use of experimental and archaeological tools. It was applied to a collection of archaeological sickles from several archaeological sites dating from the Natufian to the Late Pre-Pottery Neolithic B periods in the Near East [37–40].

A recent paper [41] demonstrated the capacity of quantitative texture analysis of 3D images obtained through confocal microscopy to consistently identify experimental tools used for working different contact materials (bone, antler, wood, fresh hide, dry hide, wild cereals, domestic cereals, and reeds). That study represented a considerable advance towards the aim of establishing a quantitative methodology for the identification of worked materials in use-wear analysis. However, most of the experimental tools correctly classified in that study were used for relatively long periods of time (sixty minutes or more). It was therefore not clear how efficient the method is for correctly classifying experimental tools used for shorter periods of time. Thus, previous models for microwear quantification through texture analysis are static, as they only consider the type of worked material, despite the influence of the intensity of use is widely recognized as a crucial factor conditioning micropolish texture [3, 16].

This paper aims to show how microwear polish evolves over time and how polish development affects our capacity of identification of worked materials using texture analysis and confocal microscopy. The dynamic nature of use-wear polish is taken into account, by analyzing how its texture changes through time of use. We test the discriminating capacity of the method to correctly classify use-wear polishes at different stages of development. We show that polish formation is a dynamic process, although most of the texture modification takes place during the first phases of development. We demonstrate that microwear polish caused by different worked materials evolves in different ways. As a result, despite the dynamic nature of polish formation, it is possible to identify worked materials for most of the stages of polish development, at least, up to one hour of use. However, errors in determination and indeterminations can appear during the first stages of development. Similarly, some degree of overlapping between pairs of materials (i.e. bone/antler) exists in well-developed polishes (40 to 60 minutes of use). These problems in the identification are caused by the dynamic nature of polish, which continues evolving in texture during the whole time of use, even if the main textural changes take place in the first steps of polish development. We show that these problems can be overcome in good measure by characterizing the use-wear polish of different worked materials in different durations of use (i.e. 10–20 minutes; 30–40 minutes; 50–60 minutes).

## Materials and methods

### The experiments

Ten experimental tools, corresponding to twelve active zones (Figs 1 and 2), were used sequentially to work six different contact materials, for 10, 20, 30, 40, 50 and 60 minutes, for a total of 72 experiments of 10 minutes of duration each (Figs 3 and 4). About 60–80 strokes per minute were made in each experiment, for a total of 3,600–4,800 strokes for each experiment. All of them were made on the same flint type, a fine-grained flint from Upper Cretaceous formations in the Donbass region (Ukraine).

- Antler. Two tools (Fig 1A and 1B) were used to scrape antler. Both tools were hand-held, wrapped in a leather wrap to prevent hand damage. As active zones, a retouched edge (80˚)

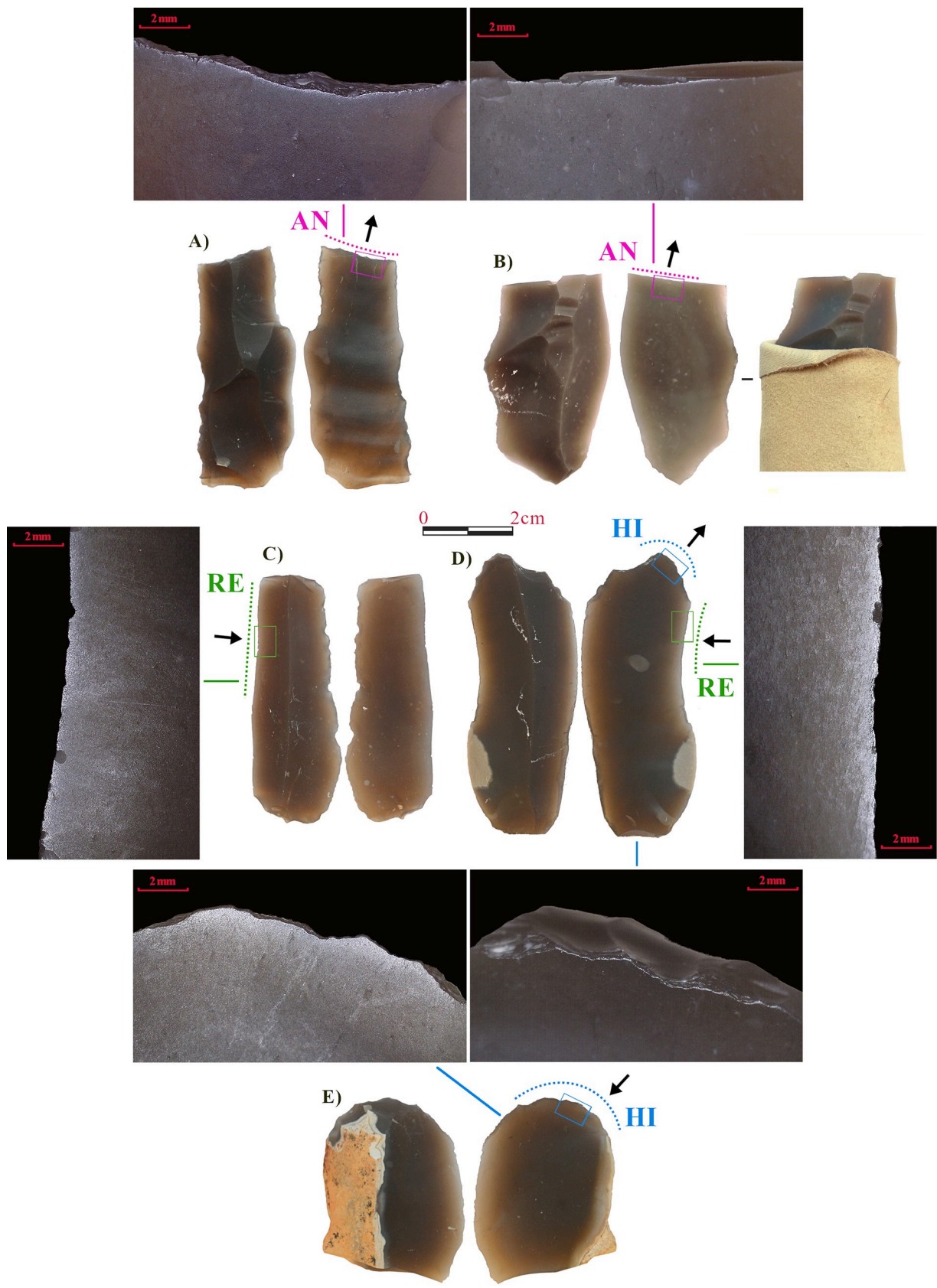

**Fig 1. Experimental tools and associated use-wear traces.** A-B) Tools used for scraping antler; C-D) Tools used for scraping reeds; D-E) Tools used for scraping hide.

was chosen for the first tool (A), while a fresh fracture with a steep angle (90˚) was used in the second one (B). Both tools were used for scraping with an outward direction. Antler was previously soaked in water for 24h, and it was regularly wetted during the use.

- Reed. Two tools (Fig 1C and 1D) were used to scrape fresh reeds, recently gathered. Both tools were hand-held. Unretouched active zones characterized by acute edges (both 30˚) were used. Tools were used with an inward movement, to remove fibers from the reed surface.

- Hide. Two tools (Fig 1D and 1E) were used to scrape dry hide (wild boar) in order to remove the epidermis. Edges were previously retouched to create an endscraper, with a steep working angle (70˚). One of them was used with an inward movement (D) and the other with an outward direction. Both were hafted in a wooden haft (E).

- Fresh bone. One tool with two active zones was used (Fig 2A). The tool was fitted into a wooden haft and used to scrape fresh bone. The two active zones, two steep natural fractures (100˚), were both used with an inbound movement. The bone surface was occasionally soaked with water during working.

- Dry wood. One tool used on two different edges (Fig 2B) to scrape dry pine wood (Fig 2A and 2B). The first edge is the distal fracture (90˚) used to scrape with an inbound movement, hand-held; the second one is the proximal portion of the left edge (60˚), also unretouched, used with an inbound movement and fitted in a wooden haft.

- Cured meat. Two tools were used to remove thin slices from a piece of cured meat. A pig hind leg was salted and left to cure for a year. Both tools were used unhafted. The first tool is a small blade used on the proximal portion (Fig 2C), an acute unretouched edge (20˚), with an inward movement. The second tool is a large flake (Fig 2D), used on the right distal edge, also unretouched and acute (20˚), with an inward movement.

## Methods

In every step, tools were cleaned with soapy water in an ultrasonic tank and the working edge was scanned with the Sensofar Plu Neox white light scanning confocal microscope, using a X20 (0.45 NA) objective, with a spatial sampling of 0.83 μm, optical resolution of 0.31 μm, vertical resolution of 20 nm and a z-step interval of 1 μm. The stitching system provided by the microscope was used in order to scan an extended surface of the used edge. Areas of $4 \times 1.5$ mm were scanned in order to choose the most adequate areas for the analysis. Samples of $50 \times 50$ μm were selected from the stitched areas. The size of the samples was chosen because antler or bone working tools do not show extended polished surfaces, so it was not possible to choose more extensive areas for this contact material and we aimed to maintain the size of the analyzed surface constant for all the contact materials. Two strategies of sampling were used in this analysis. In the first, the most developed areas (from here on MDA) were selected, whatever their position. In a second strategy (from here on 4D) we decided to sample the same place on the edge during the six steps of use. For this, the 4D module provided by Mountains7 was used; it allows the stacking of a series of measurements of the same surface. This procedure ensures a point-to-point matching of surfaces using automatic and manual shifting tools, so it is possible to apply preprocessing operators and parameters of texture measurement

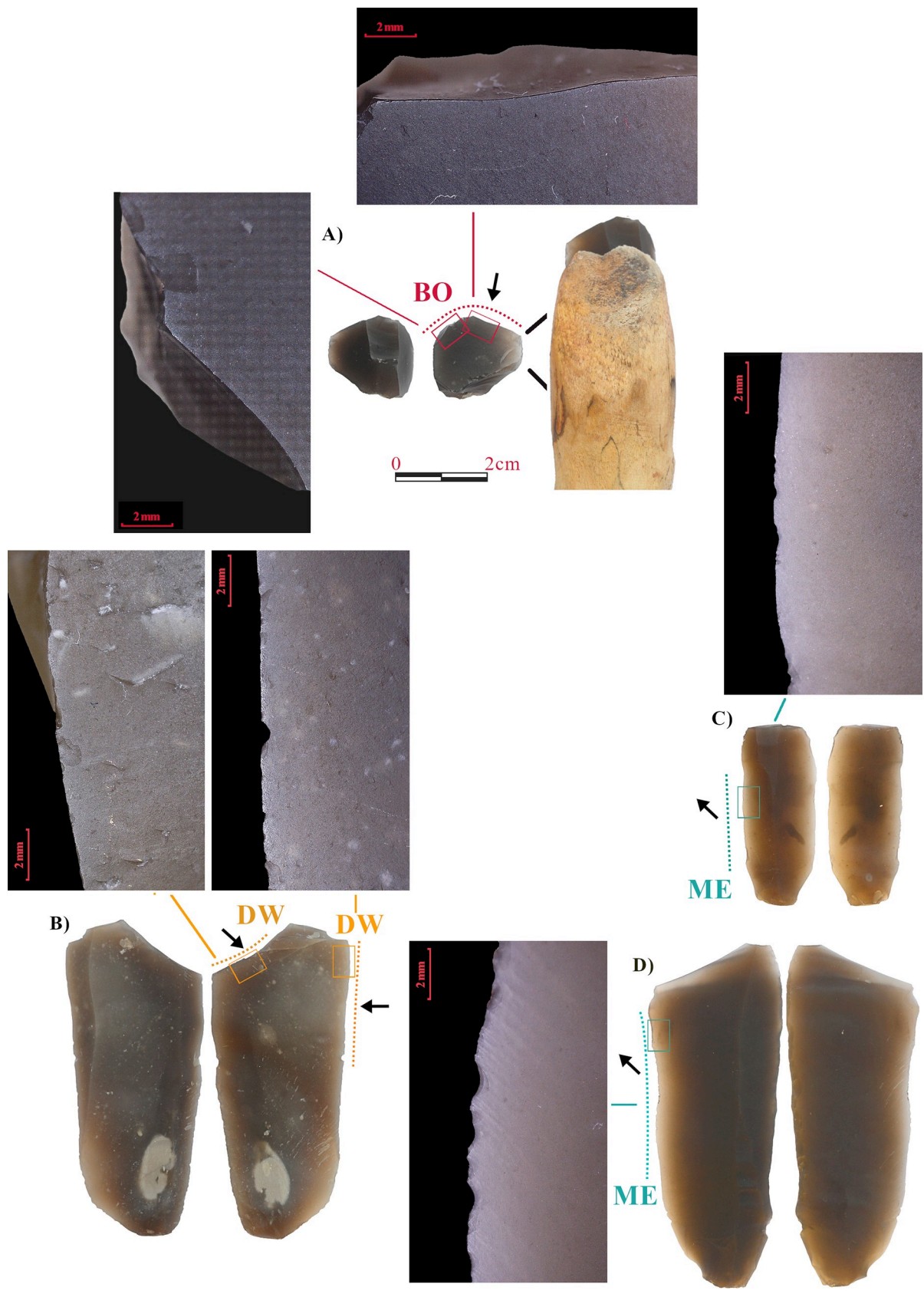

**Fig 2. Experimental tools and associated use-wear traces.** A) Tool used for scraping fresh bone, two active zones; B) Tool used for scraping dry wood, two active edges; C-D) Tools used for cutting cured meat.

simultaneously to the series. This is done by defining 50 × 50 μm areas cutting through the stacked surfaces, so exactly the same area is measured for the unused state and the six steps of use of each tool (Fig 5). The first sampling strategy (MDA) was used to test the capacity of the method for the correct classification of the worked material at different steps of development (from 10 to 60 minutes). MDA was used for this task because it mimics the procedure of selection of the most developed areas of use-wear polish that an analyst follows when he/she studies the edge of a tool. The second procedure (4D) was used to analyze the evolution of polish development over time of use in the same point of the edge. In this way, the textural characteristics of the flint surface stay the same, so the evolution of texture at different steps of use (from 10 to 60 minutes) due to the friction with the worked material can be monitored in detail.

For the first sampling strategy (MDA), samples of 50 × 50 μm of natural (unpolished) flint surfaces were obtained for each tool, totaling 117 images. 16 images of 50 × 50 μm were further sampled in each step of use of each experimental tool. In this way, 16 samples per step of use multiplied by 6 steps of use (from 10, 20, 30, 40, 50 and 60 minutes) amount to 96 samples per tool, which, multiplied for the 12 tools (two for antler, reeds, hide, bone, wood and cured meat) and adding the unpolished surfaces, resulted in 1,269 samples (S1-0 in S1 Data). For 4D sampling, eight areas were sampled for each tool, so these 8 areas multiplied by 7 steps of use (unused, 10, 20, 30, 40, 50, and 60) and by 12 tools totaled 672 samples (S2-0 in S2 Data).

These samples were processed and later measured with the Mountains7 software from Digital Surf. The processing of samples before measuring was intended to correct for the lack of horizontality of the surface. For this, a levelling operator using the least squares (LS) plane method was used. Processing was also used to separate polish texture from the irregularities of the flint surface, using a spatial filtering, which is done by moving a small filtering matrix (called a kernel matrix) over the surface. The arithmetic mean operator consists of averaging each point with its 13 × 13 neighboring points. The texture, which is the surface measured in our analysis, is calculated by subtracting the filtered surface from the source surface. Texture parameters included in ISO 25178 were used for the analysis. Among them, we chose the combination of parameters offering better discriminatory capacity through quadratic discriminant function analysis of the samples corresponding to the unused, antler, reed, hide, bone, wood and meat groups. We tested the texture parameters offered by Mountains7, choosing those that passed the tolerance test and showed statistical significance in the tests of group means (Wilks' Lamba; S2 Data). This procedure was able to select 22 parameters (Table 1).

Quadratic discriminant function analysis was used to treat the data, building a predictive model for group membership, which is composed of discriminant functions based on quadratic combinations of predictor variables when these variables show different variance–covariance matrices. The classification rule of the predictive analysis is based on Bayes' theorem.

## Results

The first question that we aimed to answer was: How does polish from different contact materials (antler, reed, hide, bone, wood, and meat) evolve over time? For this, we first measured the evolution of five texture parameters (Sq, Sdq, Sdr, Sk, Smc) during the time of use using the information provided by the 4D sampling (Fig 6) (S2-0 in S2 Data). We calculated the regression function explaining the evolution of data from the unused surface to 60 minutes of use. Curve fitting was tested through ANOVA (S2-1 in S2 Data). As result, for the six worked

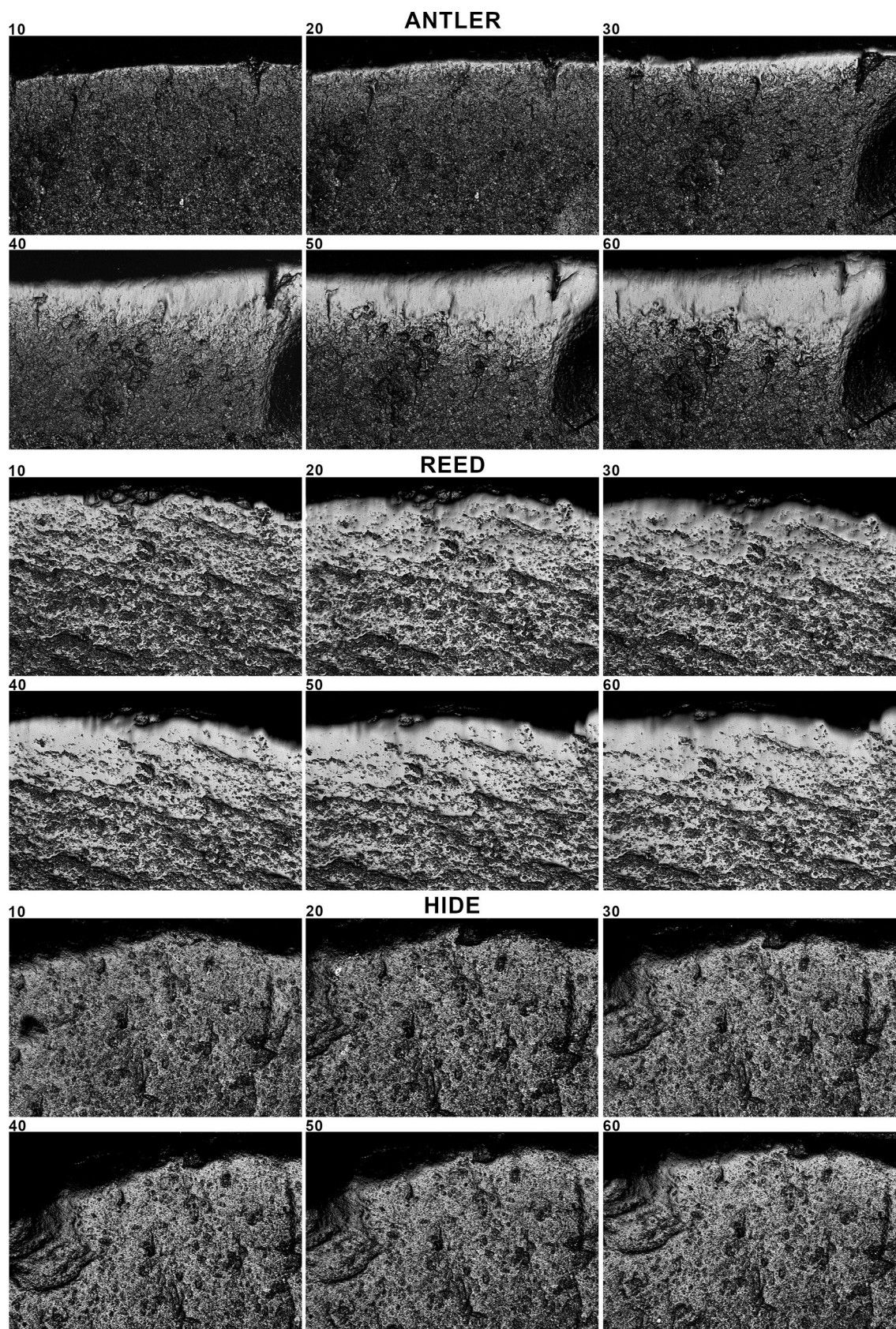

**Fig 3. Micrographs of the use-wear traces taken of the same spot at different time intervals (10, 20, 30, 40, 50, 60 minutes) for antler, reed and hide scraping.** Micrographs taken at 200× with the Sensofar Plu Neox white light scanning confocal microscope.

materials, the logarithmic function showed a good fit (F value > 0.000) (Fig 6) (S2-1 in S2 Data).

Polish texture evolves during all the steps of polishing, up to 60 minutes. The evolution of these five parameters globally indicates a mean reduction in surface roughness, a texture becoming flatter, with a reduction in surface slopes, and a reduction in the height of the core surface. In general, alterations are more important during the first steps, while the intensity of change tends to slow down after 20 or 30 minutes of use. For antler, reed and hide more than the 50% of the shift in the value of each parameter is reached at 20 minutes' use (S2-2 in S2 Data). Reed, though causing significant regularization of the surface during the first 20 minutes of use, shows a more marked and continuous change in texture up to 60 minutes of working time. For bone, wood, and especially meat, surface change is more gradual, and a marked decrease in values is reached only after 40 minutes (S2-2 in S2 Data).

The dynamic nature of microwear polish can also be observed by running a quadratic discriminant analysis, using four time intervals as grouping variable (i.e. 0, 10–20, 30–40, 50–60 minutes) (Fig 7). Results show a coherent distribution, despite some minor incongruences. Centroids move away from the unused surface as the time of use increases. The major increase is always between unused and used surface (10–20 minutes). For reed and antler, centroids are more clearly separated at each time interval. Conversely, hide, bone, wood and meat, show lesser distance between the centroids of the last two-time intervals, confirming a certain stability in traces after 30–40 minutes of working time. Meat centroids are less clearly organized, suggesting minor changes in surface texture across the time period, from 10 to 60 minutes.

These data suggest that polish development is a dynamic process, at least for the period of use considered in this work (1 hour). The degree of modification of the surface from the unmodified condition is more abrupt in the first steps of use. After a certain time of use the surface tends to be modified less intensively, defining a logarithmic model of surface modification. However, the characteristics of this model depend on the type of worked material. Reed and hide polish alter the unmodified surface faster than the other contact materials, while wood and meat are the materials showing a slower evolution of texture parameters.

The second question to be answered was: Can worked materials be correctly identified at different stages of development? To shed light on this question, we used the MDA surfaces and omitted the 4D ones, as MDA better replicates the way in which an analyst would choose the more characteristic polished areas in archaeological tools. We ran the quadratic discriminant analysis for the unused surfaces and for all the stages of development of antler, reed, hide, bone, wood and meat working. Results show a good degree of correct classification. More than 66% of the samples are well discriminated (Table 2). Wilk's Lambda analysis shows good significance for the six classificatory functions and the test of equality of group means is significant for all the textural parameters used (S1-1 in S1 Data). More than 70% of the samples of reed, hide, wood and meat working are well classified. This is also the case for more than 50% of samples of reed and 40% of bone. The two main problems of overlapping observed are the unused samples that are wrongly classified as meat-cutting in 48% of the cases and bone samples that are classified as antler in 23% of cases. The problem of overlapping is especially important for meat-cutting and unused surfaces, as more samples of unused surfaces are misclassified as meat-cutting than correctly classified as unused. In fact, meat-cutting samples are correctly classified, but many unused surfaces show similar characteristics to those surfaces used for meat-cutting.

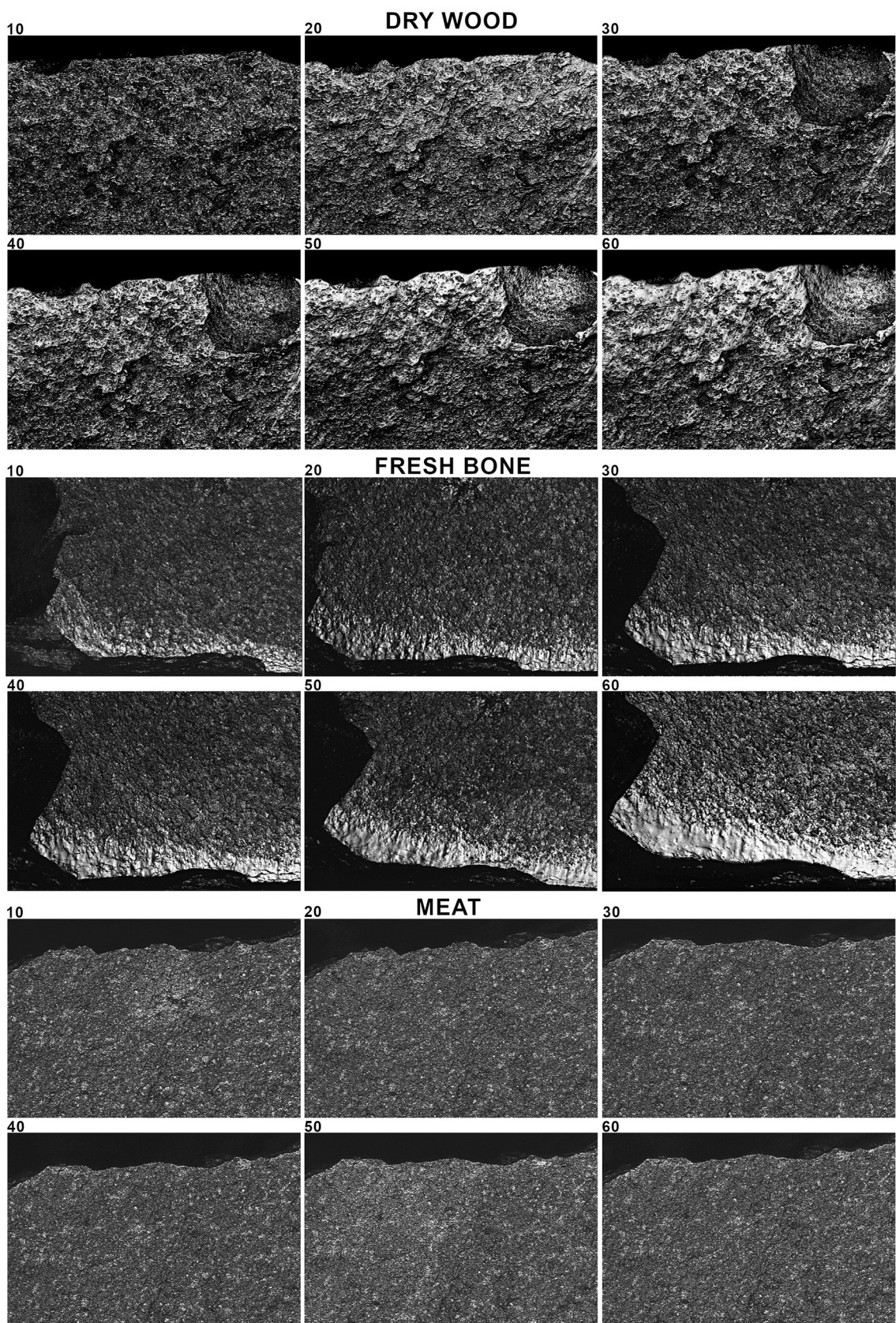

**Fig 4. Micrographs of the use-wear traces taken of the same spot at different time intervals (10, 20, 30, 40, 50, 60 minutes) for dry wood and fresh bone scraping and meat cutting.** Micrographs taken at 200× with the Sensofar Plu Neox white light scanning confocal microscope.

A further test was therefore made by exclusively considering the meat-cutting and the unused surfaces. As results, meat-cutting surfaces are well classified (86.6%), while unused surfaces clearly overlap with meat-cutting ones (56.4%) (Table 3). This suggests that the textural variability of unused surfaces can match the traces of meat-cutting tools. However, we have to bear in mind that the samples of unused surfaces were obtained from all the experimental tools. Therefore, a series of 30 unused areas from only the tools used for meat-cutting was selected (S1-2 in S1 Data). These areas were used to classify the meat-cutting areas, in order to test whether it is possible to distinguish meat-cutting traces from the natural surfaces of the same used tools. In this way, we tried to avoid the variability of flint texture corresponding to other tools than the meat-cutting ones. The classification of unused and surfaces used for meat-cutting in Experiment 1 yields a good rate of correct classification (80% for unused surfaces and almost 92% for used surfaces), with good discriminatory ability of the function (Wilk's lambda) (S1-3 in S1 Data). The classification of Experiment 2 of meat-cutting is also successful (almost 94% correct classification of unused surfaces and the same proportion of used surfaces) and with good discriminatory ability of the function (Wilk's lambda) (S1-4 in S1 Data).

These results suggest that meat-cutting traces can be distinguished from unused areas in one tool. However, the texture of one type of fine-grained flint is variable enough to overlap

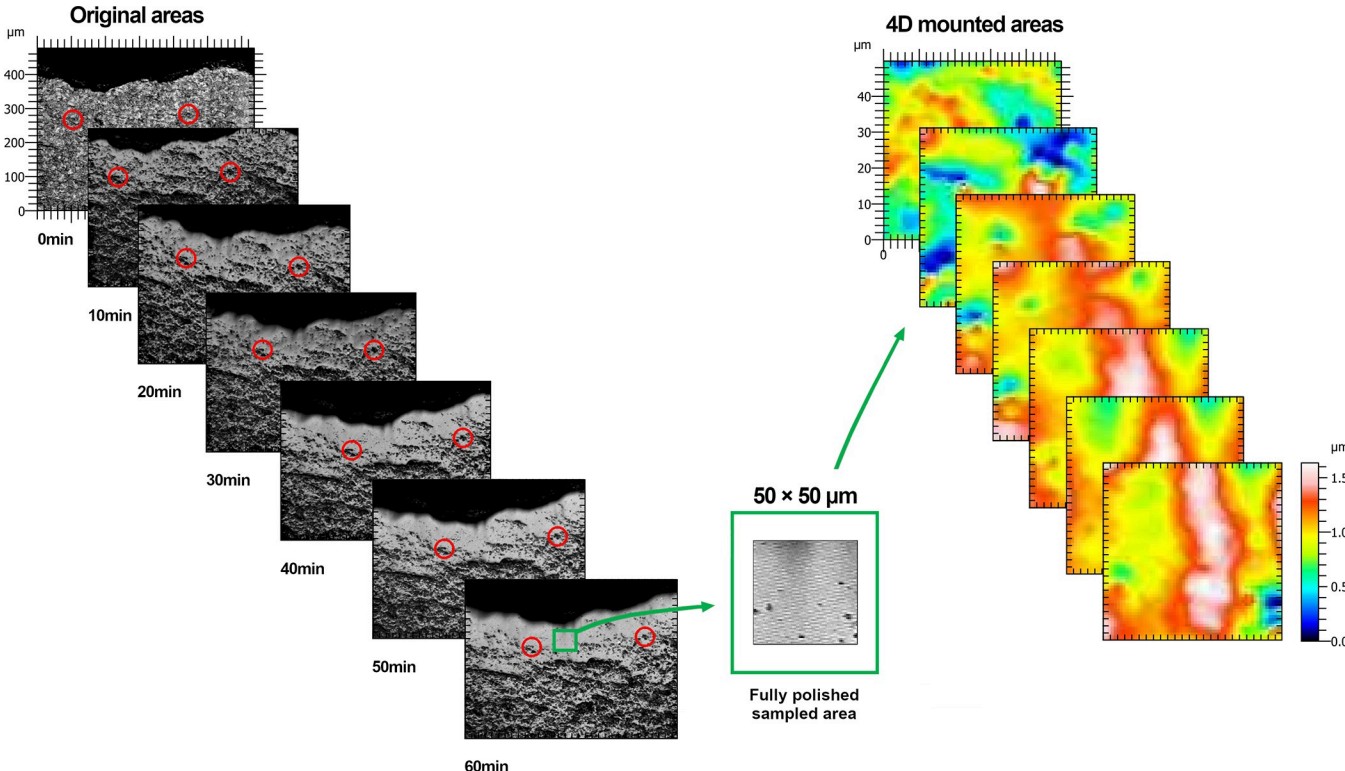

**Fig 5.** 4D sampling method: a) Example of how areas were stacked from two known points (indicated by the red dots). b) Full polished areas mounted from 10 to 60 minutes.

**Table 1. Parameters of texture used in the analysis.**

| Acronym | Name | Description |
|---|---|---|
| Sq | Root mean square height | The standard deviation of the height distribution |
| Sv | Maximum pit height | Depth between the mean plane and the deepest valley. |
| Smc | Inverse areal material ratio | Height c at which a given areal material ratio p is satisfied. The height is calculated from the mean plane |
| Str | Texture-aspect ratio | The ratio of the shortest decrease length at 0.2 from the autocorrelation, on the greatest length. This parameter has a result between 0 and 1. If the value is near 1, we can say that the surface is isotropic, i.e. has the same characteristics in all directions. If the value is near 0, the surface is anisotropic, i.e. has an oriented and/or periodical structure. |
| Sdq | Root-Mean-Square slope of the surface | The root-mean-square value of the surface slope |
| Sdr | Developed interfacial area ratio | The developed surface indicates the complexity of the surface thanks to the comparison of the curvilinear surface and the support surface. A completely flat surface will have an Sdr near 0%. A complex surface will have an Sdr of some percents. |
| Vm | Material volume | Volume of the material at a material ratio p (10%). |
| Vvv | Pit void volume of the scale limited surface | Volume of void in the valleys, between a material ratio p (80%) and 100% material ratio, calculated in the zone below c2 |
| Spd | Density of peaks | Number of peaks per unit area. |
| Spc | Arithmetic mean peak curvature | Arithmetic mean of the principle curvatures of peaks within a definition area. This parameter can determine the mean form of the peaks: either pointed, either rounded, according to the mean value of the curvature of the surface at these points. |
| S10z | Ten point height | Average value of the heights of the five peaks with the largest global peak height added to the average value of the heights of the five pits with the largest global pit height, within the definition area. |
| S5p | Five point peak height | Average value of the heights of the five peaks with the largest global peak height, within the definition area. |
| Sda | Closed dale area | Average area of dales connected to the edge at height c. |
| Sdv | Closed dale volume | Average volume of dales connected to the edge at height c |
| Shv | Closed hill volume | Average volume of hills connected to the edge at height c. |
| Sk | Kernel roughness depth | Roughness depth of the core |
| Svk | Reduced valley depth | Roughness depth of the valleys |
| Smr1 | Upper material ratio | Indicates the percentage of material that comprises the peak structures |
| Smr2 | Lower material ratio | The measurement area that comprises the deeper valley structures |
| Sds | Number of peaks per unit area | Peaks are detected by local neighbourhood (with respect to 8 neighbouring points) without discrimination between local and significant peaks, which differentiates it from Spd |
| Smean | Mean height of the surface | Mean height of the surface |
| Stdi | Texture Direction Index, | A measure of how dominant the dominating direction is, defined as the average amplitude sum divided by the amplitude sum of the dominating direction |

with meat-cutting microwear polish, resulting in the misclassification of unused surfaces, which can be wrongly identified as areas affected by friction with meat. Then, our first conclusion is that meat-cutting tools cannot be discriminated from unused surfaces following our current experimental protocol, as unused surfaces of the Donbass fine-grained flint can show a similar texture to those used for cutting meat. The identification of meat and butchery cutting tools should be approached in a specific investigation (see Discussion).

Given such results, meat-cutting samples have therefore been excluded from the classification, focusing our research on determining the classificatory capacity of the other five types of materials depending on the time of use. As a result, almost 69% of the samples of the five

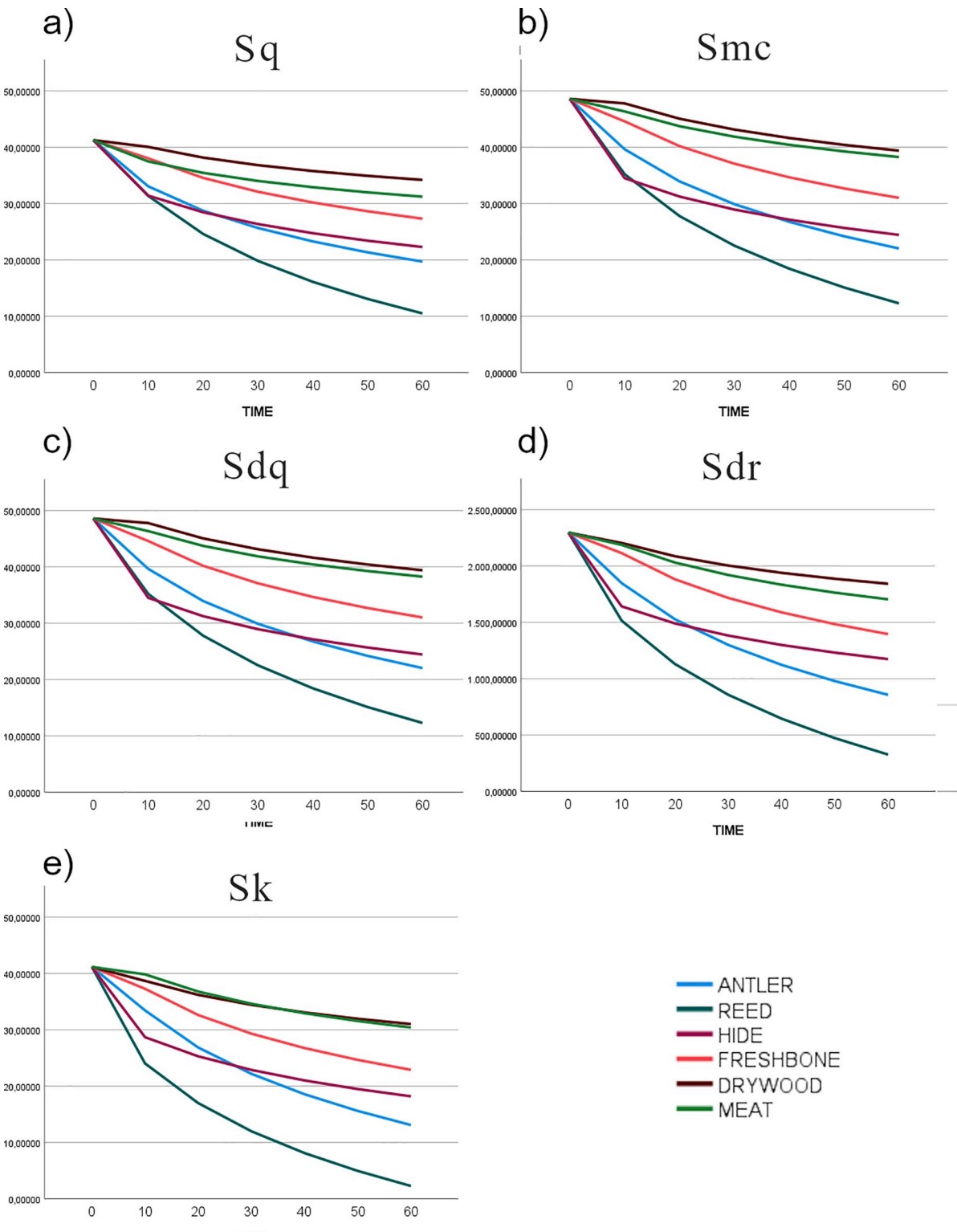

**Fig 6.** Evolution of five texture parameters Sq (a) Smc (b) Sdq (c) Sdr (d), Sk (e) through time (0, 10, 20, 30, 40, 50, 60 minutes). Predicted values fitting a logarithmic regression.

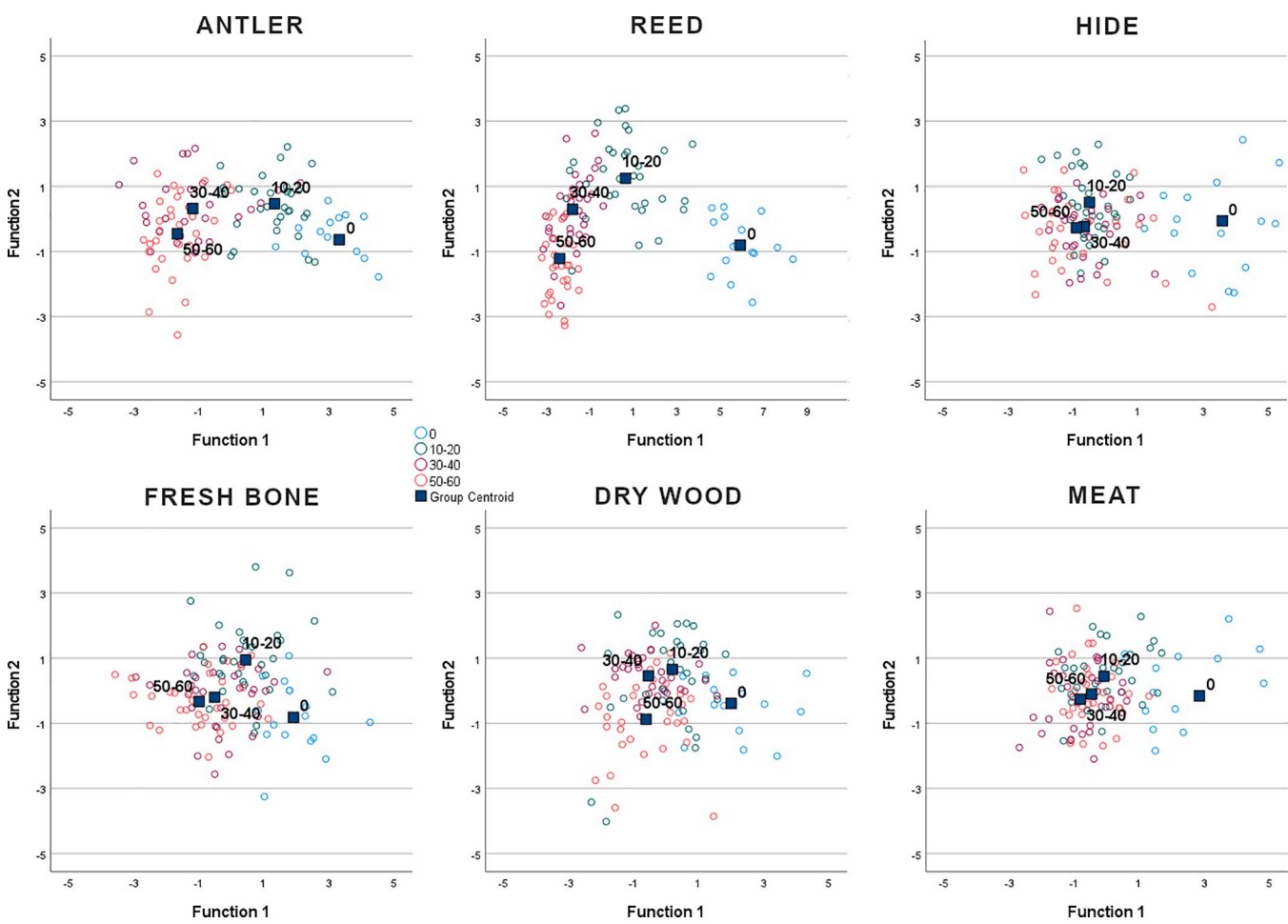

**Fig 7. Quadratic discriminant analysis of antler, reed, hide, bone wood and meat polish at 0, 10, 20, 30, 40, 50 and 60 minutes.**

worked materials (antler, bone, wood, reeds and hide) are well grouped. The proportion of correct classification for no use, wood, hide and reeds is over 70%, while for antler and bone it drops to around 50% (Table 4).

To confirm the confidence of the classification rule for each step over the time of use, we blindly classified half of the samples of each worked material every 10 minutes of use (Table 5). The blind classification consists of the grouping of the samples without previously indicating which group of worked material they belong to.

The blind test resulted in a good rate of correct classification of 60% of the samples. This rate is reached after 20 minutes' use, as, after 10 minutes' use, the proportion of correct classification is only 37.5%. However, some differences can be observed in the five worked materials. Analyzing the results, we can define the correct classification when:

- $\geq$ 50% of the samples are correctly classified and < 30% of the samples are grouped in a second erroneous group;

- 40% to 50% of the samples are correctly classified and < 20% of the samples are grouped in a second erroneous group.

In the opposite case (classification of a greater number of samples in a wrong group), the classification should be considered as erroneous. When the difference between the quantity of

**Table 2. Results of classification of the six worked materials (antler, reed, hide, bone, wood and meat) and no use by quadratic discriminant analysis.** 66.6% of original grouped cases were correctly classified. Shaded cells correspond to correct classifications.

| | | **Classification Results** | | | | | | | |
|---|---|---|---|---|---|---|---|---|---|
| | | **Predicted Group Membership** | | | | | | | |
| | | **UNUSED** | **ANTLER** | **REED** | **HIDE** | **BONE** | **WOOD** | **MEAT** | **Total** |
| Count | UNUSED | 47 | 5 | 0 | 4 | 1 | 4 | 56 | 117 |
| | ANTLER | 5 | 100 | 24 | 21 | 9 | 4 | 29 | 192 |
| | REED | 0 | 12 | 161 | 8 | 9 | 2 | 0 | 192 |
| | HIDE | 0 | 23 | 3 | 143 | 12 | 5 | 6 | 192 |
| | FRESH BONE | 3 | 45 | 15 | 20 | 86 | 20 | 3 | 192 |
| | WOOD | 8 | 0 | 1 | 1 | 6 | 145 | 31 | 192 |
| | MEAT | 17 | 3 | 0 | 1 | 1 | 12 | 175 | 209 |
| % | UNUSED | 40,2 | 4,3 | 0 | 3,4 | 0,9 | 3,4 | 47,9 | 100 |
| | ANTLER | 2,6 | 52,1 | 12,5 | 10,9 | 4,7 | 2,1 | 15,1 | 100 |
| | REED | 0 | 6,3 | 83,9 | 4,2 | 4,7 | 1 | 0 | 100 |
| | HIDE | 0 | 12 | 1,6 | 74,5 | 6,3 | 2,6 | 3,1 | 100 |
| | BONE | 1,6 | 23,4 | 7,8 | 10,4 | 44,8 | 10,4 | 1,6 | 100 |
| | WOOD | 4,2 | 0 | 0,5 | 0,5 | 3,1 | 75,5 | 16,1 | 100 |
| | MEAT | 8,1 | 1,4 | 0 | 0,5 | 0,5 | 5,7 | 83,7 | 100 |

samples in the first group and the second is not so clear-cut and there are two or three groups with similar results, a second analysis can be carried out to better define the classification rule, reducing the number of groups to only the two or three main groups concerned in the analysis. This can happen when:

- ≥ 50% of the samples are classified in one group and > 30% of the samples are grouped in a second one;

- 40% to 50% of the samples are classified in one group and 20% to 30% of the samples are grouped in a second one.

- Two or three worked materials group between 30 and 40% of the samples, so none of them clearly stands over the others.

If with a second analysis > 60% of the samples are grouped in one category of worked material, the classification can be considered correct. Conversely, if the previous conditions are not respected in the classification, the classification can be considered indeterminate.

Following this classificatory rule, we can evaluate the results of the blind test (Table 6). At 10 minutes' use, antler samples are wrongly classified, as more than 60% of the samples are in the group of unused surfaces. After 20 minutes of working time, the classification is

**Table 3. Results of classification of meat-cutting and unused surfaces by quadratic discriminant analysis.** A high proportion of the unused surfaces (56.4%) are wrongly classified as meat-cutting. Shaded cells indicate the correct classification.

| | | **Classification Results—Meat cutting** | | |
|---|---|---|---|---|
| | | **Predicted Group Membership** | | |
| | | **UNUSED** | **MEAT** | **Total** |
| Count | UNUSED | 51 | 66 | 117 |
| | MEAT | 28 | 181 | 192 |
| % | UNUSED | 43,6 | 56,4 | 100 |
| | MEAT | 13,4 | 86,6 | 100 |

**Table 4. Results of classification of the five worked materials (antler, reed, hide, bone and wood) and no use by quadratic discriminant analysis.** 68.5% of original grouped cases were correctly classified. Shaded cells indicate the correct classification.

| | | Classification Results | | | | | | |
|---|---|---|---|---|---|---|---|---|
| | | Predicted Group Membership | | | | | | |
| | | UNUSED | ANTLER | REED | HIDE | BONE | WOOD | Total |
| Count | UNUSED | 100 | 6 | 0 | 4 | 2 | 5 | 117 |
| | ANTLER | 29 | 100 | 23 | 26 | 12 | 2 | 192 |
| | REED | 0 | 14 | 163 | 7 | 5 | 3 | 192 |
| | HIDE | 7 | 24 | 2 | 142 | 10 | 7 | 192 |
| | BONE | 3 | 39 | 14 | 18 | 91 | 27 | 192 |
| | WOOD | 38 | 3 | 0 | 1 | 8 | 142 | 192 |
| % | UNUSED | 85,5 | 5,1 | 0 | 3,4 | 1,7 | 4,3 | 100 |
| | ANTLER | 15,1 | 52,1 | 12 | 13,5 | 6,3 | 1 | 100 |
| | REED | 0 | 7,3 | 84,9 | 3,6 | 2,6 | 1,6 | 100 |
| | HIDE | 3,6 | 12,5 | 1 | 74 | 5,2 | 3,6 | 100 |
| | BONE | 1,6 | 20,3 | 7,3 | 9,4 | 47,4 | 14,1 | 100 |
| | WOOD | 19,8 | 1,6 | 0 | 0,5 | 4,2 | 74 | 100 |

indeterminate, with all the categories grouping less than 40% of samples. From 30 to 60 minutes' working time, the classification is correct. At 50 minutes of work, more than 30% of the samples are erroneously grouped as reeds, but a second classification between antler and reeds offers a high proportion of correct classification (81.3%). Reeds classification is indeterminate at 10 minutes, while it starts to be correct from 20 minutes and the correct grouping continues up to 60 minutes. Hide-working is correctly classified during the six steps of development, from 10 to 60 minutes. Bone-working is more difficult to identify. The classification at 10 and 20 minutes is indeterminate. After 10 minutes, a second analysis between wood and bone, the two groups with most samples, results in the classification of 50% of the samples in each group. At 20 minutes, the same proportion of 31.3% of the samples are grouped as hide, bone and wood. A second analysis groups 25% of the samples as hide, 43.8% as bone and 31.3% as wood, so the result is indeterminate. After 30 and 40 minutes, bone samples are well grouped, though at 50 and 60 minutes they are mixed with antler. After 50 minutes, a similar proportion of bone samples are classified as antler and bone. If a second classification grouping the bone samples between either bone or antler categories is tried, the results are not conclusive, as only 56.3% of the samples are correctly classified as bone. Thus, for this time interval (50 minutes) the identification would be either bone or antler. Bone is also mixed with antler at 60 minutes, as half of the samples are wrongly classified as antler and only 31% as bone. If a second classification is carried out between the two categories, 81.3% of the samples are correctly grouped as bone. Finally, wood-working tools are correctly classified even from 10 minutes of use, up to 60 minutes. However, at 50 minutes, wood samples are partially mixed with bone. Even if 50% of samples are identified as wood, more than 30% are misclassified as bone. A second classification between the two categories does not resolve the indetermination, as only 59% of the samples are correctly classified as wood.

One of the two wood-scraping experiments was carried out with a hand-held tool and the other with a hafted tool. The development of use-wear polish was more intense and rapid for the hafted tool. Thus, for the same time of use, the polish in the hafted tool was clearly better developed. To check how this difference (hafted or unhafted) can affect the degree of development of the polish and our capacity to identify the worked material, we analyzed the classification of both tools separately (S1-5 in S1 Data). While the classification of the unhafted tool is

**Table 5. Results of classification of the five worked materials (antler, reed, hide, bone and wood) every 10 minutes of use by quadratic discriminant analysis.** Half of the surface samples (n = 8) of each group of worked material/time of use (i.e. bone 10 minutes, hide 20 minutes. . .) were blindly classified in the five groups of worked materials plus no use. Shaded cells indicate the correct classification.

| Time | Material | | NO USE | ANTLER | REEDS | HIDE | BONE | WOOD | Total |
|---|---|---|---|---|---|---|---|---|---|
| | | | MDA BLIND CLASSIFICATION (EVERY 10 MINUTES) | | | | | | |
| 0 | UNUSED | Σ | 100 | 6 | - | 4 | 2 | 5 | 117 |
| | | % | 85,5 | 5,1 | - | 3,4 | 1,7 | 4,3 | 100% |
| 10 | ANTLER | Σ | 10 | 1 | - | 3 | 1 | 1 | 16 |
| | | % | 62,5 | 6,3 | - | 18,8 | 6,3 | 6,3 | 100% |
| | REED | Σ | - | 3 | 4 | 4 | 3 | 2 | 16 |
| | | % | - | 18,8 | 25 | 25 | 18,8 | 12,5 | 100% |
| | HIDE | Σ | 3 | 2 | - | 10 | - | 1 | 16 |
| | | % | 18,8 | 12,5 | - | 62,5 | - | 6,3 | 100% |
| | BONE | Σ | - | 2 | - | 3 | 4 | 7 | 16 |
| | | % | - | 12,5 | - | 18,8 | 25 | 43,8 | 100% |
| | WOOD | Σ | 4 | 1 | - | - | - | 11 | 16 |
| | | % | 25 | 6,3 | - | - | - | 68,8 | 100% |
| 20 | ANTLER | Σ | 4 | 3 | - | 5 | 1 | 3 | 16 |
| | | % | 25 | 18,8 | - | 31,3 | 6,3 | 18,8 | 100% |
| | REED | Σ | - | - | 15 | - | 1 | - | 16 |
| | | % | - | - | 93,8 | - | 6,3 | - | 100% |
| | HIDE | Σ | - | 2 | - | 14 | - | - | 16 |
| | | % | - | 12,5 | - | 87,5 | - | - | 100% |
| | BONE | Σ | - | 1 | - | 5 | 5 | 5 | 16 |
| | | % | - | 6,3 | - | 31,3 | 31,3 | 31,3 | 100% |
| | WOOD | Σ | 5 | - | - | - | - | 11 | 16 |
| | | % | 31,3 | - | - | - | - | 68,8 | 100% |
| 30 | ANTLER | Σ | - | 11 | - | 3 | 1 | 1 | 16 |
| | | % | - | 68,8 | - | 18,8 | 6,3 | 6,3 | 100% |
| | REED | Σ | - | 2 | 14 | - | - | - | 16 |
| | | % | - | 12,5 | 87,5 | - | - | - | 100% |
| | HIDE | Σ | - | 2 | - | 13 | 1 | - | 16 |
| | | % | - | 12,5 | - | 81,3 | 6,3 | - | 100% |
| | BONE | Σ | 1 | 2 | - | 2 | 9 | 2 | 16 |
| | | % | 6,3 | 12,5 | - | 12,5 | 56,3 | 12,5 | 100% |
| | WOOD | Σ | 6 | - | - | - | - | 10 | 16 |
| | | % | 37,5 | - | - | - | - | 62,5 | 100% |
| 40 | ANTLER | Σ | - | 9 | 3 | 3 | 1 | - | 16 |
| | | % | - | 56,3 | 18,8 | 18,8 | 6,3 | - | 100% |
| | REED | Σ | - | - | 14 | 1 | 1 | - | 16 |
| | | % | - | - | 87,5 | 6,3 | 6,3 | - | 100% |
| | HIDE | Σ | 1 | 4 | - | 9 | 1 | 1 | 16 |
| | | % | 6,3 | 25 | - | 56,3 | 6,3 | 6,3 | 100% |
| | BONE | Σ | 1 | 2 | - | 2 | 9 | 2 | 16 |
| | | % | 6,3 | 12,5 | - | 12,5 | 56,3 | 12,5 | 100% |
| | WOOD | Σ | 1 | - | - | - | - | 15 | 16 |
| | | % | 6,3 | - | - | - | - | 93,8 | 100% |

(*Continued*)

**Table 5.** (Continued)

| | | | NO USE | ANTLER | REEDS | HIDE | BONE | WOOD | Total |
|---|---|---|---|---|---|---|---|---|---|
| **Time** | **Material** | | | | | | | | |
| **50** | **ANTLER** | Σ | 1 | 8 | 5 | - | 2 | - | 16 |
| | | % | 6,3 | 50 | 31,3 | - | 12,5 | - | 100% |
| | **REED** | Σ | - | 3 | 13 | - | - | - | 16 |
| | | % | - | 18,8 | 81,3 | - | - | - | 100% |
| | **HIDE** | Σ | 1 | 3 | 2 | 8 | 1 | 1 | 16 |
| | | % | 6,3 | 18,8 | 12,5 | 50 | 6,3 | 6,3 | 100% |
| | **BONE** | Σ | - | 6 | 2 | 3 | 5 | - | 16 |
| | | % | - | 37,5 | 12,5 | 18,8 | 31,3 | - | 100% |
| | **WOOD** | Σ | 1 | 1 | 1 | - | 5 | 8 | 16 |
| | | % | 6,3 | 6,3 | 6,3 | - | 31,3 | 50 | 100% |
| **60** | **ANTLER** | Σ | - | 9 | 4 | - | 3 | - | 16 |
| | | % | - | 56,3 | 25 | - | 18,8 | - | 100% |
| | **REED** | Σ | - | - | 16 | - | - | - | 16 |
| | | % | - | - | 100 | - | - | - | 100% |
| | **HIDE** | Σ | - | 2 | 1 | 11 | 1 | 1 | 16 |
| | | % | - | 12,5 | 6,3 | 68,8 | 6,3 | 6,3 | 100% |
| | **BONE** | Σ | - | 8 | 2 | - | 6 | - | 16 |
| | | % | - | 50 | 12,5 | - | 37,5 | - | 100% |
| | **WOOD** | Σ | - | 1 | - | - | 3 | 12 | 16 |
| | | % | - | 6,3 | - | - | 18,8 | 75 | 100% |

*MDA BLIND CLASSIFICATION (EVERY 10 MINUTES)*

problematic at 10 and 20 minutes, with a high proportion of samples classified as unused, the identification of the worked material for the hafted tool is correct from 10 minutes of use. However, the identification of the worked material for the hafted tool is problematic for the advanced phases of use (50 and 60 minutes), when wood polish partially overlaps with bone.

In sum, the results obtained from the classification of the five worked materials through time of use indicate that, the main problems in the identification of the worked material appear during two phases of use:

- The first phases, from 0 to 20 minutes: The difficulties affect the harder and more rigid materials, such as antler and bone (for 10 and 20 minutes) and reeds (for 10 minutes). For antler, after 10 minutes the difficulty is to discriminate the used areas from unused ones. For the rest (antler 10 minutes, reeds 10 minutes, bone 10 and 20 minutes), the identification as indeterminate is the result of a disperse classification of samples, in which samples are attributed to several worked materials.

**Table 6. Results of the blind classification of experimental tools per time of use when all the phases of use (from 10 to 60 minutes) are classified together in the five worked materials.**

| | Working Time | | | | | |
|---|---|---|---|---|---|---|
| **MAT** | **10** | **20** | **30** | **40** | **50** | **60** |
| **Antler** | Error | Indeterminate | Correct | Correct | Correct in second analysis | Correct |
| **Reed** | Indeterminate | Correct | Correct | Correct | Correct | Correct |
| **Hide** | Correct | Correct | Correct | Correct | Correct | Correct |
| **Bone** | Indeterminate | Indeterminate | Correct | Correct | Bone/Antler | Correct in second analysis |
| **Wood** | Correct | Correct | Correct | Correct | Wood/Bone | Correct |

- The advanced phases of use (50 and 60 minutes): At these phases there is relative overlapping between pairs of worked materials, such as bone with antler (50 and 60 minutes), antler with reeds (50 minutes) and wood with bone (50 minutes). This overlapping provokes the need to resort to a second analysis between two categories of worked materials that in some cases results in correct classification (antler 50 minutes and bone 60 minutes) or in binary classifications (bone or antler for bone 50 minutes and wood or bone for bone 50 minutes).

These results suggest that, even if use-wear polish is a dynamic phenomenon and its characteristics change over time, discriminant texture analysis of 3D images obtained through confocal microscopy is solid enough to obtain good classificatory results for the five worked materials from 10 to 60 minutes of use. However, following this procedure, some problems of classification appear during the first phases of use, especially for the harder and more rigid materials, and during the more advanced phases of use. At those initial and advanced phases of use, a relative level of indetermination appears in the classification. Therefore, it seems that the dynamic nature of polish entails that, if all the phases of use (from 10 to 60 minutes) are classified together for the five worked materials, a risk of indetermination or even error appears for the first and the more advanced phases of development.

Since the dynamic nature of polish introduces uncertainty in the classification of the worked material, we have tested a new classification taking account, at the same time, the worked material and the time of use. For this, we have run the quadratic discriminant analysis of the samples by grouping them in 16 categories instead of five. These categories were the result of crossing the worked materials with four phases of use: 0 minutes, 10–20 minutes, 30–40 minutes and 50–60 minutes. Considering these 16 categories, 70% of the samples are correctly classified (S1-6 in S1 Data). The test of equality of groups' means is significant for all the predictor variables.

Again, to test the predictive capacity of the classificatory rules, we blindly classified half of the samples of each group of worked material every 10 minutes of use, classifying them in the 16 groups of worked material/time plus no-use (S1-7 in S1 Data). To evaluate the capacity of this test to identify the contact material, we show the results by contact material, adding the results of the three time intervals (Table 7).

Following our rule of classification, all the phases of antler working are correctly classified (Table 8). Antler overlaps with hide during 20 and 30 minutes, and a second analysis is needed to clearly discriminate the samples between them. In both cases, 62.6% of the samples are correctly classified as antler. Reed and hide samples are correctly classified from 10 to 60 minutes. Bone 10 minutes is correctly classified in a second classification against wood, with 62.5% of the samples grouped in the former category. After 20 and 30 minutes, bone working is correctly classified. At 40 minutes, 43.75% of the samples are wrongly classified as antler and 25% as bone. As these rates fall in the conditions needing a second analysis, the discriminant analysis was carried out between bone and antler tools. Only 56.4% of the samples are grouped as bone, so the identification of antler at 40 minutes would be binary: either bone or antler. At 50 minutes, bone working is also mixed with antler, but a second discriminant analysis between the two worked materials results in the correct classification as bone of 62.5% of the samples. Wood working was correctly classified from 10 to 60 minutes.

Plotting the disposition of the centroids of the first two classificatory functions for the 16 groups (Fig 8; including 73.1% of cumulative variance), we can observe that each worked material is placed in a specific area. However, the evolution of texture through time for each worked material implicates changes in the position of the centroids.

A further test has been carried out in order to determine how hafting affects the development of the traces and, consequently, our capacity to identify the worked material. We

**Table 7. Summary by worked material of the results of classification of the five worked materials (antler, reed, hide, bone and wood) every 10 minutes of use by quadratic discriminant analysis.** Half of the surface samples (n = 16) of each group of worked material/time of use (i.e. bone 10 minutes, hide 20 minutes. . .) were blindly classified in 16 groups of worked materials and four intervals of time of use (0, 10–20, 30–40, 50–60 minutes). Shaded cells indicate the correct classification.

| | Predicted Group Membership in % | | | | | |
|---|---|---|---|---|---|---|
| | No use | Antler | Reeds | Hide | Bone | Wood |
| Antler 10 | 18,8 | 43,8 | - | 18,8 | 6,3 | 12,6 |
| Antler 20 | 12,5 | 31,25 | - | 31,25 | 6,25 | 18,75 |
| Antler 30 | - | 50 | - | 37,5 | 12,5 | - |
| Antler 40 | - | 81,25 | 12,5 | - | 6,25 | - |
| Antler 50 | - | 68,75 | 12,5 | 12,5 | 6,25 | - |
| Antler 60 | - | 68,75 | 12,5 | - | 18,75 | - |
| Reed 10 | - | 12,5 | 43,75 | 12,5 | 18,75 | 12,5 |
| Reed 20 | - | 6,25 | 75 | - | 18,75 | - |
| Reed 30 | - | 12,5 | 75 | 6,25 | 6,25 | - |
| Reed 40 | - | 6,25 | 87,5 | 6,25 | - | - |
| Reed 50 | - | 31,25 | 68,75 | - | - | - |
| Reed 60 | - | - | 100 | - | - | - |
| Hide 10 | 6,25 | 25 | - | 68,75 | - | - |
| Hide 20 | - | - | - | 100 | - | - |
| Hide 30 | - | 6,25 | - | 87,5 | - | 6,25 |
| Hide 40 | - | 18,75 | - | 62,5 | 12,5 | 6,25 |
| Hide 50 | - | 31,25 | 6,25 | 62,5 | - | - |
| Hide 60 | - | 25 | 6,25 | 5- | 18,75 | - |
| Bone 10 | - | 6,25 | - | 12,5 | 50 | 31,25 |
| Bone 20 | 6,25 | 6,25 | 6,25 | 12,5 | 62,5 | 6,25 |
| Bone 30 | - | 18,75 | 18,75 | 6,25 | 50 | 6,25 |
| Bone 40 | - | 43,75 | 12,5 | 12,5 | 25 | 6,25 |
| Bone 50 | - | 31,25 | 18,75 | 6,25 | 43,75 | - |
| Bone 60 | - | 25 | 6,25 | 12,5 | 56,25 | - |
| Wood 10 | 25 | 6,25 | - | - | 12,5 | 56,25 |
| Wood 20 | - | 25 | - | - | - | 75 |
| Wood 30 | - | 18,75 | - | - | 6,25 | 75 |
| Wood 40 | - | 31,25 | 12,5 | 6,25 | - | 50 |
| Wood 50 | 6,25 | 18,75 | - | - | - | 75 |
| Wood 60 | - | - | - | 6,25 | 6,25 | 87,5 |

classified independently the two tools used for wood working, one of which was used hafted while the other was held in the bare hand. For the hand-held tool, problems of classification are evident after 10 minutes of working time (overlapping with no use), and 20, 40 and 50

**Table 8. Results of the blind classification of experimental tools per time of use when all the phases of use (from 10 to 60 minutes) are classified in the five worked materials and four intervals of use (0, 10–20 minutes, 30–40 minutes, 50–60 minutes).**

| | Working Time | | | | | |
|---|---|---|---|---|---|---|
| MAT | 10 | 20 | 30 | 40 | 50 | 60 |
| Antler | Correct | Correct in second analysis | Correct in second analysis | Correct | Correct | Correct |
| Reed | Correct | Correct | Correct | Correct | Correct | Correct |
| Hide | Correct | Correct | Correct | Correct | Correct | Correct |
| Bone | Correct in second analysis | Correct | Correct | Bone/Antler | Correct in second analysis | Correct |
| Wood | Correct | Correct | Correct | Correct | Correct | Correct |

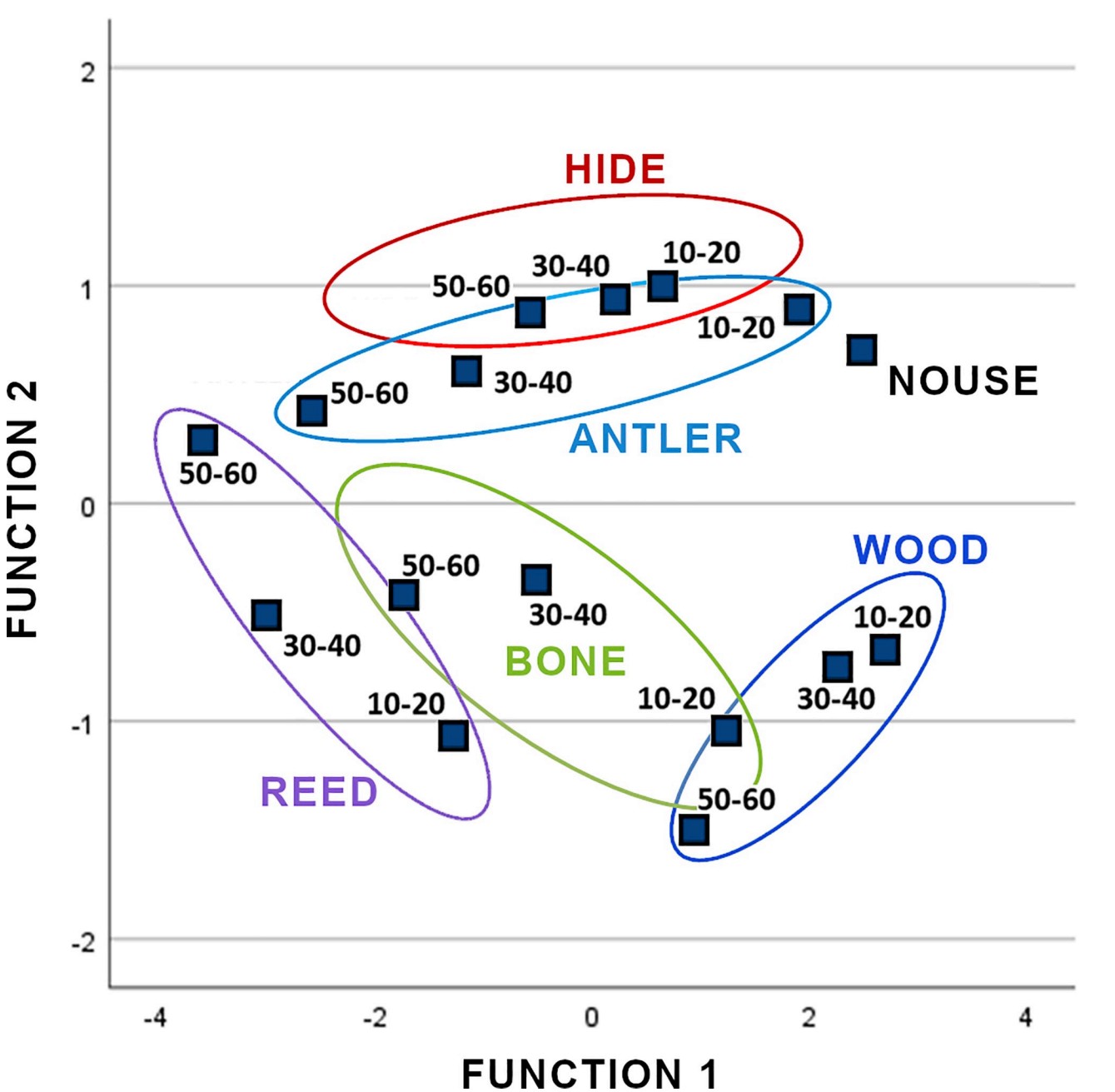

**Fig 8. Plot with the two best functions of the quadratic discriminant analysis.** Points represent centroids of the 16 groups of worked materials/time of use plus unused surfaces. Function 1 explains 62.6% of the variance and Function 2 10.6%.

minutes (overlapping with antler). In a second classification between wood and no use for the samples used for 10 minutes, the classification is still ambiguous (50% for no use and 50% for wood). The second classification between wood and antler for 20, 40 and 50 minutes yields correct classifications (S1-6a, S1-6b in S1 Data). Conversely, the classification of the hafted tool against the 16 groups (material/time plus no use) is correct for woodworking from 10 to 60 minutes. These results indicate that the hafted tool develops more polish, resulting in better classification of the tool (S1-7 in S1 Data). In conclusion, considering both the time of use and the worked material, a better identification of the worked material is reached.

This classification also offers indications on the degree of trace development. To test this, we calculated an index of degree of development of polish based on data given in Table 7, in which we considered the samples correctly classified for each worked material. Among them, we summed the proportion of samples identified in the three time-intervals (10–20, 30–40, 50–60), also including samples classified as unused. Then the resulting proportion was multiplied by 1 for the results of the 10–20 interval, by 2 for the second interval (30–40), and by 3 for the third (50–60); each number was finally divided by 3. This means that if all samples in each group (i.e. Antler 10 minutes, bone 20 minutes. . .) were classified in the 10–20 working time, the corresponding index would be 33, while if all samples were classified in the 50–60 minutes group, the index would be 100. The graphs (Fig 9) show that, although minor incongruences occur, there is a constant increase in trace development through the three time intervals (10–20, 30–40, 50–60). However, the obtained values often fall below or exceed the expected proportion. When the index exceeds the expected value for the specific time-interval, it means that some samples have been erroneously classified with samples from longer working times; conversely, when the value is lower than the expected proportion it means that traces are less developed than the expected model.

## Discussion

Quantitative use-wear analysis is a growing field of research. Already over 30 years ago, several scholars, using different methodologies, demonstrated that polish generated by diverse worked materials could be quantitatively discriminated [21–24, 28, 41–44]. During the last decade, confocal microscopy and texture analysis have been used for the same aim, showing the great potential of this methodology [30, 31, 5–47] for tools made with different rocks, including non-homogeneous ones, like quartzite [48]. Confocal microscopy has taken this approach one step forward, as it is a precise and easy-to-use method for the analysis of textures. It has been possible to confidently classify experimental tools depending on the worked material [47] and the method has been applied to collections of archaeological tools [37–39].

Despite such advances, many challenges remain and confocal microscopy should be developed further to test its potentiality and limits. Each new study explores new issues related to different facets of this novel technique: the area to be measured and sampled, the magnification to be used, the most relevant surface parameters, etc. Use-wear from different worked materials (plant, animal) [37, 39, 41] and on different raw -materials (flint, pottery, bones, etc.) [35, 48, 49, 51, 52] are tested and gradually included in this field of study. All these pilot studies will build a consensus on practices to be followed.

In this sense, one of the big issues in relation to use-wear traces that remained unexplored is related to their dynamic nature. How does polish produced by different worked materials evolve over time? How does polish development affect the capacity of discriminating the worked materials quantitatively? Is it possible to approach polish development through quantitative techniques?

Data obtained in this study, using sequential experimentation [50–53] and 4D documentation of used surfaces, have provided relevant information on polish development. The analysis of the evolution of five texture parameters (i.e. Sq, Smc, Sdq, Sdr, and Sk) and the quadratic discriminant analysis of tools used at different stages of use showed that:

1. Surface polishing is a dynamic process and texture evolves continuously up to 60 minutes of work.

2. During use, flint surfaces become smoother, with a reduction in surface roughness, flatter texture, and a reduction in surface slopes.

## POLISH DEVELOPMENT THROUGH TIME

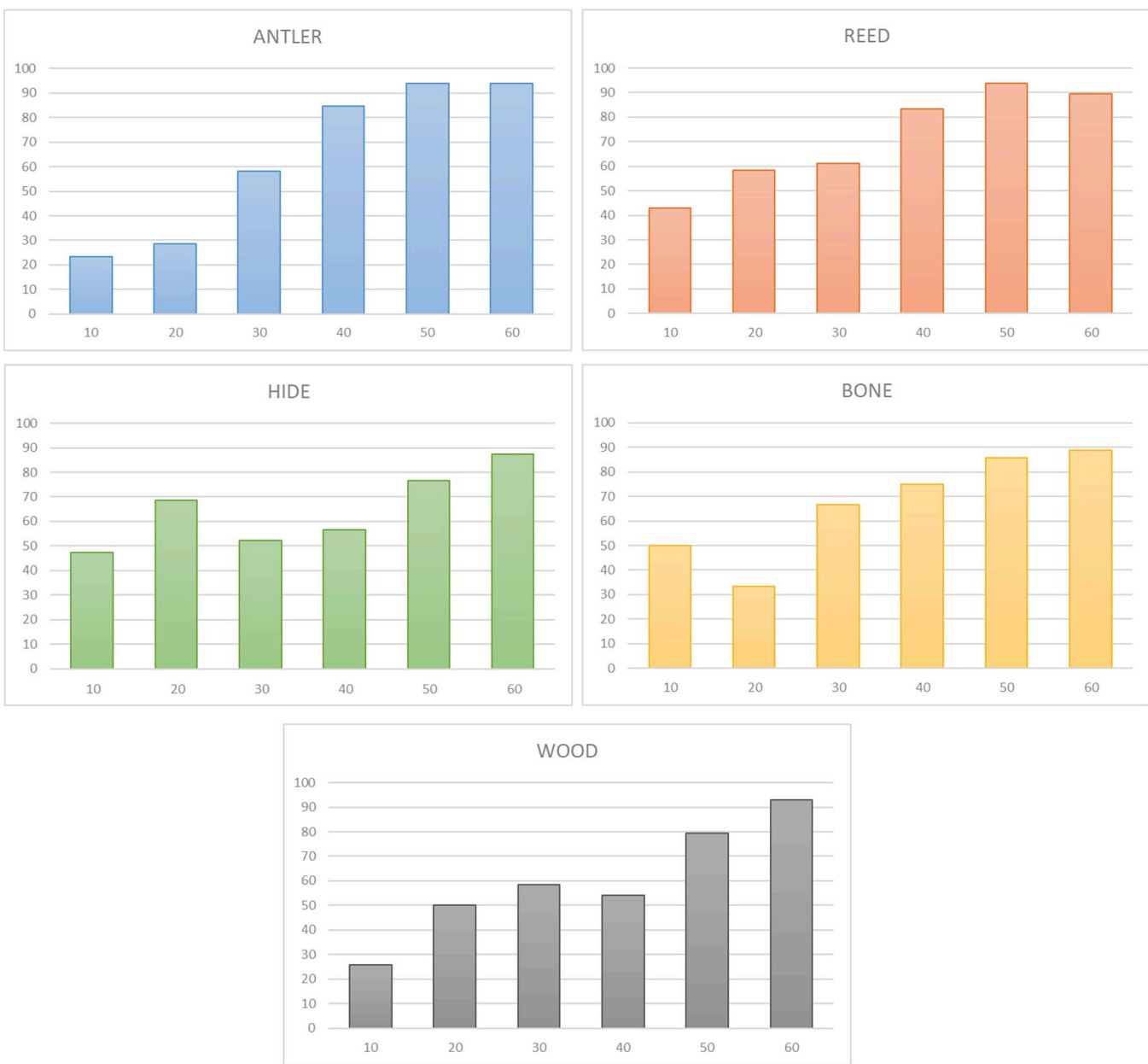

**Fig 9. Histograms showing the index of polish development predicted by the quadratic discriminant analysis for the groups of worked materials (antler, reed, hide, bone and wood) and time of use (10, 20, 30, 40, 50, 60 minutes).** Half of the surface samples (n = 8) of each group of worked material/time of use (i.e. bone 10 minutes, hide 20 minutes. . .) were blindly classified in the 16 groups (i.e. no use, wood 10–20 minutes, hide 30–40 minutes. . .). Among the results of the predicted group membership, those in which the worked material was correctly identified or were grouped as no use were retained. The number of surface samples identified as no use were multiplied by 0; those identified in the group 10–20 minutes of use were multiplied by 1; those identified in the group 30–40 minutes of use were multiplied by 2; those identified in the group 50–60 minutes of use were multiplied by 3. After adding the previous quantities, the result was divided by 3. Each result of this calculation for each group of worked material/ time of use (i.e. bone 10 minutes, hide 20 minutes. . .) is plotted in the columns of the histogram.

3. This evolution fits a logarithmic function, so a large part of the textural transformation takes place during the first stages of polish development. After these first stages of development, texture alters less intensively.

4. The evolution of polish texture is different for the five worked materials.

Some of these characteristics of polish evolution were suggested in previous studies of polish quantification. González-Urquijo and Ibáñez [24], using image analysis, observed that the degree of linkage of bone, hide, plant and antler polish (polish pattern) evolved over time of use following an exponential regression curve. Each type of polish displayed differences in its evolution. Considering the same step of development, the polish types were organized from less to more in the following order: bone, antler, hide and plants. Stemp and Stemp [54], using UBM laser profilometry, observed that the surface roughness values for pottery-sawing flint tools were different at sequential intervals of use. However, they did not manage to document the evolution of texture through time of use in another experimental tool used for cutting wood. Evans et al. [55] observed that antler polish changed in texture through time of use, but they were not able to test a similar evolution for a woodworking tool. We have observed changes in texture for all the worked materials although, interestingly, wood polish is the one that changes the least, which could explain the results obtained by Stemp and Stemp and Evans et al.

The classification of the experimental tools used sequentially allowed some conclusions to be reached about the capacity of the quantitative method to identify the worked material at different times of use. It was difficult to discriminate meat-cutting traces from the unused surfaces as was the case in another recent study using Laser scanner Confocal Microscopy [56]. The variability in texture of the flint surface overlaps with the meat-cutting traces. Our results suggest that it is possible to distinguish meat-cutting polish from the unused areas of the same tool. However, taking together the unused surfaces of our experimental tools that were made from different nodules of the same fine-grained flint, their textural variability overlaps with meat-cutting traces. The characterization of meat cutting and butchery traces will be specifically dealt with in future research. Thus, our analysis concentrated on the tools used with antler, reeds, hide, bone and wood.

We observed that it is possible to identify the worked material in experimental tools used from 10 to 60 minutes. However, we found some level of uncertainty and even error in the identification of the worked material in the first phases of use (10 and 20 minutes), especially for the harder and more rigid materials, as bone, antler and reeds. We also found some overlapping between some worked materials in the more advanced phases of use (40 to 60 minutes), when the use-wear polish is well developed, like for antler/bone or bone/wood. Our quantitative results concur with the experience of dozens of specialists who have been working in use-wear analysis during the last four decades. General experience indicates that identification is problematic during the first phases of polish development. However, after a certain time of use, microwear polish reaches a phase of stabilization after which the specific characteristics of the modified surface allow the identification of the worked material [2–7, 57]. It is also generally accepted that there is some overlapping in polish patterns between well-developed polishes [3, 58, 59], for example, between antler and bone, wood and antler, wood and plants and so on.

These problems in the identification of the worked materials by texture analysis can be mostly overcome by creating models of polish texture in which not only the worked materials but also the time of use are considered. The blind classification of the experimental tools in 16 groups, including the five worked materials (antler, reeds, hide, bone and wood) in three stages of time of use (10–20, 30–40 and 50–60 minutes) plus unused surfaces offers a better rate of correct classification than just classifying the tools in the five groups of worked materials. Thus, future models of polish texture should consider both the worked material and the time of use. Moreover, this classification taking account of the worked material and the time of use

can provide some information on the intensity of use. In this paper, we have controlled the intensity of use by measuring the time of use, considering that our work was regular through time. However, it is not possible to identify the time of use in experimental tools of unknown use or in archaeological tools, as factors other than the time of use affects the intensity of the traces. For example, in this paper, we have shown that the hafted woodworking tool displayed better developed polish than the tool used in the bare hand. However, our research shows that it is possible to obtain information on the intensity of the surface modification in a specific area, which can be useful for comparing intensities of use among different active areas of the same tool or even between different tools.

What have we learnt in this study compared with previous models of polish development obtained through qualitative methods? Use-wear polish is a dynamic process over time of use, which confirms R. Grace's claims. However, most of the textural modifications take place during the first phases of use, while in the more advanced ones polish is more stable, which can be associated with the third phase of polish development advocated by P. Vaughan. However, polish from working different materials does not pass through the same phases of texture modification as defended by R. Grace. The existence of different 'ways of development' of polish generated by the worked materials explains why experienced use-wear analysts are able to identify the worked material by qualitative methods, as the human brain can build visual models in which polish from different worked materials is characterized in the different steps of use. If the qualitative method is valid, do we really need quantitative methods? In general, quantitative methods provide more reliable identifications, as they are not biased by inter- and intra-analyst inconsistencies. Quantitative data on use-wear can be stocked, shared and models continuously improved with new experiments. Moreover, as we have shown in our analysis of sickle gloss, quantitative methods allow more detailed identifications (i.e. wild cereals harvested in natural stands, cultivated wild cereals, domestic cereals, reeds and other grasses) which are very difficult even for experienced specialists. Discussing the role of quantitative methods in use-wear analysis merits detailed argumentation in a new paper. In any event, quantitative and qualitative methods are definitively complementary.

Although the dynamic nature of polish was assumed by use-wear analysts, no models of polish evolution have been proposed, apart from P. Vaughan's very general three categories. This lack of models explains two weaknesses of qualitative methods regarding the identification of worked materials: 1) the capacity to discriminate the worked materials in the first phases of use (generic weak polish) and 2) the overlapping of different types of polishes in the more advanced phases of development (i.e. bone/antler or bone/wood). These two difficulties have been quantitatively demonstrated in our study. For the first time, we have defined specific models of polish development for the five worked materials up to one hour of work. Our quantitative models combining type of worked material and phase of development allow these limitations to be partially overcome. First, it is possible to discriminate the five worked materials in the first phases of use, at least after 10 minutes of work. Second, in our models, overlapping of worked materials is restricted to antler/bone between 40 and 50 minutes of work.

We have shown that the analysis of polish texture allows the identification of the intensity of use. This opens a new field of research that should be considered with care. We have used the time of use as the vector defining the intensity of work. However, at the same time, we have shown that hafted and unhafted tools used for the same lapse of time result in different intensities of polish development. Other factors besides hafting, such as the strength or experience of the worker, the way in which the worked material is held (i.e. a hide in a frame or on a surface) and so on, can equally affect the intensity of use. This question opens a new challenge for quantitative use-wear analysis. If the time of work is not the only factor affecting the intensity of use (and the texture modification of polish), how can we build precise dynamic models

of polish development? For example, a specific degree of textural modification might correspond to working for 50 minutes with an unhafted tool or for 15 minutes with a hafted one. We have defined the intensity of use as steps of 10 minutes of work. Most probably, future dynamic models of use-wear polish will be defined as multifactorial regressions of textural parameters, so tools of unknown intensity of use would be identified with a relative point of intensity of work in the regression model. In any case, polish texture should be used to compare different intensities of use only IF the rest of the variables characterizing work are similar (i.e. sickles used for harvesting).

All in all, this study opens new perspectives on the understanding and modeling of use-wear polish formation process. Quantitative analysis enables numerical comparison of visual aspects, such as smoother / rougher surfaces. The understanding of how surface texture parameters evolve through time of use improves our comprehension of wear formation, and the rate of surface modification for different types of worked materials. In the near future, it will be necessary to address the identification of the activities of cutting soft animal tissues and butchery. The implementation of additional experimental programs including other classes of substances (fresh hide, fresh wood, plants other than reeds, mineral materials. . .) will provide a more comprehensive view of how use-wear traces are generated and their textural differences. Sequential experiments carried out over different times of use could indeed provide the classificatory rules for future inferential models, opening the door to the creation of an experimental reference collection for the quantitative identification of use-wear polish at different development stages.

Despite such advances, the application of the method to archaeological collections is still challenging. Some of the aspects which need further research are shared with qualitative use-wear analysis, such as the need for more realistic experiments reproducing the enormous variability of working procedures in the prehistoric past. Two of these aspects are especially relevant: the influence of the variability of the rocks used for the tools [60, 61] and the effect of post-depositional alterations [51, 62]. Regarding the first problem, we observed that the variability among different types of fine-grained flint does not affect texture analysis substantially [36] except for the identification of meat-cutting activities. However, the analysis of coarse-grained flints and other rocks will require the development of specific experimental programs [48]. As regards post-depositional alterations, our previous analysis of archaeological tools was carried out on well preserved Neolithic glossed tools [37–39]. However, the analysis of more altered archaeological collections should be preceded by precise knowledge on how post-depositional modifications affect our capacity to identify worked materials quantitatively through the study of polish texture. Other issues that should be addressed are specific to quantitative studies. In order to standardize analysis, it is necessary for specialists to reach a consensus on the technical aspects of the methodology (microscope proprieties and chosen techniques of analysis, magnifications, filtering procedure, parameters of texture and so on). Thus, we are convinced that the further development of quantitative methods implies the collaboration of several research groups.

## Conclusions

The degree of use-wear polish development affects the interpretation of the use traces [3, 42]. In this paper, the analysis of the evolution of texture parameters through time of use has allowed us to gain crucial information on polish development and how it affects the identification of the worked material. Our data demonstrate that polish development is a dynamic process, at least up to 60 minutes of use. The model of texture regularization fits a logarithmic regression curve, in which the wear process accelerates rapidly at first and then slows over

time. The polish generated by diverse worked materials evolves differently during the time of use. Even if polish texture alters over time, worked materials can be quantitatively identified. However, some overlapping appears in the first stages of development between used and unused surfaces and between worked materials. Some overlapping is also present in well-developed polishes generated by materials with similar characteristics (i.e. bone and antler). This overlapping can, in good measure, be solved by creating quantitative models in which not only the worked material but the time of use is considered.

The proposed approach represents a valuable contribution to research on use-wear polish formation and discrimination. It demonstrates that use-wear polish can be visually identified even if it is dynamic in nature, as most of the textural change takes place in the first stages of development and each worked material evolves differently over time. In this way, this study represents a quantitative demonstration supporting the analysis carried out through visual comparison of experimental and archaeological polishes. However, our analysis also confirms the limitations of the method (identification of weakly developed polishes and the overlapping between well-developed similar polishes, i.e. bone and antler). Most importantly, this study opens the door for the development of a quantitative method for the identification of worked materials, as now we know how the time of use affects polish characteristics and how to over-come these limitations, creating quantitative models in which not only the worked material but the time of use are considered. Enlarging this study to other worked materials and for lon-ger use-times will enable a quantitative method to be designed for the identification of worked materials through texture analysis using confocal microscopy. Better knowledge of how tool rock-type variability and post-depositional alterations affect our capacity to identify the worked material are now the two main challenges for the application of the method to archae-ological collections.

## Supporting information

**S1 Data.** S1-0: Database with the measures of the texture parameters of the experimental tools. MDA sampling. S1-1: Test of equality (Wilk's Lambda) of the Functions and the groups means of the texture parameters resulting from the quadratic discriminant analysis. S1-2: Database with the measures of the texture parameters of unused surfaces in the experimental tools used for cutting meat. S1-3: Results of the classification of unused and used surfaces from tool 1 used to cut meat. Shaded cells indicate correct classification. S1-4: Results of the classifi-cation of unused and used surfaces from tool 2 used to cut meat. Shaded cells indicate correct classification. S1-5: Results of the classification of the hafted and unhafted tool used for work-ing wood, Shaded cells indicate correct classification. S1-6: Results of the classification of tools every 10 minutes of use (i.e. bone 10 minutes, hide 20 minutes. . .) by quadratic discriminant analysis in 16 groups of worked materials and four intervals of time of use (0, 10–20, 30–40, 50–60 minutes). Shaded cells indicate the correct classification. S1-7: Results of the blind clas-sification of tools every 10 minutes of use by quadratic discriminant analysis. Half of the sur-face samples (n = 16) of each group of worked material/time of use (i.e. bone 10 minutes, hide 20 minutes. . .) were blindly classified in 16 groups of worked materials and four intervals of time of use (0, 10–20, 30–40, 50–60 minutes). Shaded cells indicate the correct classification. S1-8: 8a: Blind classification of surfaces in worked material/time groups used for working wood with an unhafted tool during 10, 20, 30, 40, 50 and 60 minutes. 8b: Second blind classifi-cation between unused and wood for surfaces resulting from working wood during 10 minutes with an unhafted tool; blind classification between antler and wood for surfaces resulting from working wood during 20, 40 and 50 minutes with an unhafted tool. S1-9: Blind classification of surfaces in worked material/time groups used for working wood with a hafted tool during

10, 20, 30, 40, 50 and 60 minutes.
(XLSX)

**S2 Data.** S2-0: Database with the measures of the texture parameters of the experimental tools. 4D sampling. S2-1: Calculation of the curve estimation (logarithmic model) of textural parameters Sq, Smc, Sdq, Sdr and Sk at 10, 20, 30, 40, 50 and 60 minutes. S2-2: Average values of textural parameters Sq, Smc, Sdq, Sdr and Sk at 10, 20, 30, 40, 50 and 60 minutes.
(XLSX)

## Author Contributions

**Conceptualization:** Juan José Ibáñez, Niccolò Mazzucco.

**Data curation:** Juan José Ibáñez, Niccolò Mazzucco.

**Formal analysis:** Juan José Ibáñez, Niccolò Mazzucco.

**Funding acquisition:** Juan José Ibáñez, Niccolò Mazzucco.

**Investigation:** Juan José Ibáñez, Niccolò Mazzucco.

**Methodology:** Juan José Ibáñez, Niccolò Mazzucco.

**Project administration:** Juan José Ibáñez, Niccolò Mazzucco.

**Resources:** Juan José Ibáñez, Niccolò Mazzucco.

**Software:** Juan José Ibáñez, Niccolò Mazzucco.

**Supervision:** Juan José Ibáñez, Niccolò Mazzucco.

**Validation:** Juan José Ibáñez, Niccolò Mazzucco.

**Visualization:** Juan José Ibáñez, Niccolò Mazzucco.

**Writing – original draft:** Juan José Ibáñez, Niccolò Mazzucco.

**Writing – review & editing:** Juan José Ibáñez, Niccolò Mazzucco.

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
