## [Decision Letter · Decision Letter 0]

6 Jul 2021

PONE-D-21-14694

Quantitative Usewear Analysis of Stone Tools: Measuring How the Intensity of Use Affects the Identification of the Worked Material

PLOS ONE

Dear Dr. Juan José Ibáñez,

Thank you for submitting your manuscript to PLOS ONE. After careful consideration, we feel that it has merit but does not fully meet PLOS ONE’s publication criteria as it currently stands. Therefore, we invite you to submit a revised version of the manuscript that addresses the points raised during the review process.

We look forward to receiving your revised manuscript.

Kind regards,

Enza Elena Spinapolice, Ph.D

Academic Editor

PLOS ONE

Journal Requirements:

Reviewers' comments:

Reviewer's Responses to Questions

**Comments to the Author**

1. Is the manuscript technically sound, and do the data support the conclusions?

Reviewer #1: Yes

Reviewer #2: Yes

2. Has the statistical analysis been performed appropriately and rigorously? 

Reviewer #1: Yes

Reviewer #2: I Don't Know

3. Have the authors made all data underlying the findings in their manuscript fully available?

Reviewer #1: Yes

Reviewer #2: Yes

4. Is the manuscript presented in an intelligible fashion and written in standard English?

Reviewer #1: Yes

Reviewer #2: Yes

5. Review Comments to the Author

Reviewer #1: I would like to start by acknowledging the value of this contribution to the field of quantitative use-wear-analysis. The authors provide a new set of innovative experiments aimed to enlarging the actual knowledge on the possibility to identify the materials worked with flint tools at different stages of use.

This unique fact is sufficient to recommend this paper for publication on PlosOne.

Nevertheless, there are some minor issues which should be addressed by the authors before publication. All the comments are intended to improve the quality of this work, which is already notable.

I’d suggest the authors to touch on some points which are still crucial in the development of the discipline. To name a few, there is not a general consensus on the right technical (magnifications, NA, area scanned) and analytical parameters (3D analysis) to be used, whether they should be different on different raw materials or not, etc. I’d appreciate to see these topics reflected in the discussion for several reasons. First of all, it should be clear to non-experts that quantitative research is still in its infancy and there is no consensus among experts. In this way, the authors would not create false hopes to the general public by passing on an incomplete and ambiguous message.

I, of course, agree with the authors when they say that there are still issues to be solved, such as the impact that PDSMs has on interpretations, before being ready to analyze archeological collections.

Regarding presentation of data, although the tables provided here for the classification of materials are clear, I wonder why the authors did not provide matrices instead.

Another point that is worth spending some words on (in the discussion maybe?) is in my opinion the fact that the results presented in this paper do not diverge much from observations done by analysts employing qualitative parameters. If we forget for some time about numbers, and we stick to only the descriptive paragraphs focused on the description of the development of polish, I honestly see no difference from the descriptions of the first use-wear analysts (Keeley, Vaughan, Grace). On the one hand, this is very reassuring for the discipline and analysts should not feel threatened anymore by the application of quantitative methods.

On the other hand, and this is the trickiest question to be answered by us, do we really need to go to such fine details? Do we need to go on with quantitative studies? Do we need to standardize parameters in order to compare datasets?

If the answer is yes, I would like to see why this is the case and your opinion on this matter shall somehow be explicit in the text. This would hopefully trigger a scientific debate and the discipline would follow its natural course.

Reviewer #2: Very interesting and useful paper. This approach has great potential for creating a digital open access experimental database for use-wear analysis. I cannot fully judge the statistical support of the paper, but methodologically it is certainly good. It is well illustrated, well written, but the references need a check: authors are sometimes missing and initials are incorrect. A term like editores should be changed to editors.

Although the authors make reference to previous work on quantification, they could clarify better what is really new here in comparison to past conclusions about use-wear inferences. They conclude that there is overlap between for example wood and bone and bone and antler polish and mention the difficulty of distinguishing meat cutting wear. However, we have known this for decades. So, what exactly have we learned that is different from what we already know? I am not contesting the usefulness of their approach, quite the contrary, but they might push this more, and make clear why this type of quantification can bring us further or what could be the practical applicability for a larger research community. Not everyone has access to a confocal microscope.

I am also still not convinced that we can speak of a "correct identification" of polishes: yes, of course in experimental context we can, but our inferences on archaeological tool use are based on analogies. Past technologies were most likely far more complex than we can imagine and our experiments are often way too simplistic: there are several archaeological polish types for which we have yet to find an experimental equivalent, even after decades of experimental exploration.

After the first analysis the "success rate" is presented to be around 60%, if I understand the text well. The statement that after a second round of statistical analysis virtually all classifications were correct, does not entirely convince me as this is based on a combined analysis of length of use with contact material. Length of use is, as the authors themselves acknowledge, not the only factor behind intensity of polishes (strength, skill, fatigue etc).

In conclusion my advise is "accept with minor revision". Although not questioning the usefulness of the quantitative approach presented in this paper, I think the paper would benefit from a short critical appraisal of its potential and limitations in the conclusion of the paper.

6. PLOS authors have the option to publish the peer review history of their article (what does this mean?). If published, this will include your full peer review and any attached files.

Reviewer #1: No

Reviewer #2: No

---

## [Author Response · Author response to Decision Letter 0]

16 Jul 2021

We are extremely grateful to both reviewers for their valuable commentaries.

Response to Review Comments to the Author

Reviewer #1: I would like to start by acknowledging the value of this contribution to the field of quantitative use-wear-analysis. The authors provide a new set of innovative experiments aimed to enlarging the actual knowledge on the possibility to identify the materials worked with flint tools at different stages of use.

This unique fact is sufficient to recommend this paper for publication on PlosOne.

Nevertheless, there are some minor issues which should be addressed by the authors before publication. All the comments are intended to improve the quality of this work, which is already notable.

I’d suggest the authors to touch on some points which are still crucial in the development of the discipline. To name a few, there is not a general consensus on the right technical (magnifications, NA, area scanned) and analytical parameters (3D analysis) to be used, whether they should be different on different raw materials or not, etc. I’d appreciate to see these topics reflected in the discussion for several reasons. First of all, it should be clear to non-experts that quantitative research is still in its infancy and there is no consensus among experts. In this way, the authors would not create false hopes to the general public by passing on an incomplete and ambiguous message. I, of course, agree with the authors when they say that there are still issues to be solved, such as the impact that PDSMs has on interpretations, before being ready to analyze archeological collections.

A few sentences have been added in the Discussion: ‘Quantitative usewear analysis is a growing field of research. […] Each new study explores new issues related with different facets of this new technique: the area to be measured and samples, the magnification to be used, the most relevant surface parameters, etc. Usewear from different worked materials (plant, animal) and on different raw-materials (flint, pottery, bones, etc.)35,47,48,50,5 1are tested and included in this field of study. All these pilot studies will allow building a consensus on practices to be followed.’

In the Materials and Methods section, we already explicate why we chose areas of 50 x 50 μm: ‘Samples of 50 × 50 μm were selected from the stitched areas. The size of the samples was chosen because antler or bone working tools do not show extended polished surfaces, so it was not possible to choose more extensive areas for this contact material and we aimed to maintain the size of the analyzed surface constant for all the contact materials.’

However, we agree with the Reviewer, that in the future, it will be interesting to test as well other magnifications, for example using an X50 objective. This field of research is currently being built and each new case-study contributes to its development.

Regarding presentation of data, although the tables provided here for the classification of materials are clear, I wonder why the authors did not provide matrices instead.

Matrices are provided as part of supplementary materials.

Another point that is worth spending some words on (in the discussion maybe?) is in my opinion the fact that the results presented in this paper do not diverge much from observations done by analysts employing qualitative parameters. If we forget for some time about numbers, and we stick to only the descriptive paragraphs focused on the description of the development of polish, I honestly see no difference from the descriptions of the first use-wear analysts (Keeley, Vaughan, Grace). On the one hand, this is very reassuring for the discipline and analysts should not feel threatened anymore by the application of quantitative methods.

On the other hand, and this is the trickiest question to be answered by us, do we really need to go to such fine details? Do we need to go on with quantitative studies? Do we need to standardize parameters in order to compare datasets?

If the answer is yes, I would like to see why this is the case and your opinion on this matter shall somehow be explicit in the text. This would hopefully trigger a scientific debate and the discipline would follow its natural course.

A paragraph has been added: ‘What have we learnt in this analysis with respect to previous models of polish development obtained through qualitative methods? Use-wear polish is a dynamic process all along the time of use, what confirms R. Grace’s claims. However, most of the textural modifications take place during the first phases of use, while in the more advanced ones polish is more stable, what can be identified with the third phase of polish development advocated by P. Vaughan. Definitely, polish from working different materials do not pass through the same phases of texture modification as defended by R. Grace. The existence of different ‘ways of development’ of polish generated by the worked materials explains why the experienced use-wear analysts are able to identify the worked material by qualitative methods, as the human brain can build visual models in which polish from different worked materials are characterized at the different steps of use. If the qualitative method is valuable, do we really need quantitative methods? In general, quantitative methods provide more reliable identifications, as they are not biased by inter and intra analysts inconsistencies. Quantitative data on use-wear can be stocked, shared and models continuously improved with new experiments. Moreover, as we have shown in our analysis of sickle gloss, quantitative methods allow more detailed identifications (i.e. wild cereals harvested in natural stands, cultivated wild cereals, domestic cereals, reeds and other grasses) which are very difficult even for experienced specialists. Discussing the role of quantitative methods in use-wear analysis merits detailed argumentation in a new paper. In anyway, quantitative and qualitative methods are definitively complementary.’

Reviewer #2: Very interesting and useful paper. This approach has great potential for creating a digital open access experimental database for use-wear analysis. I cannot fully judge the statistical support of the paper, but methodologically it is certainly good. It is well illustrated, well written, but the references need a check: authors are sometimes missing and initials are incorrect. A term like editores should be changed to editors.

References have been corrected.

Although the authors make reference to previous work on quantification, they could clarify better what is really new here in comparison to past conclusions about use-wear inferences. They conclude that there is overlap between for example wood and bone and bone and antler polish and mention the difficulty of distinguishing meat cutting wear. However, we have known this for decades. So, what exactly have we learned that is different from what we already know? I am not contesting the usefulness of their approach, quite the contrary, but they might push this more, and make clear why this type of quantification can bring us further or what could be the practical applicability for a larger research community. Not everyone has access to a confocal microscope.

A paragraph has been added: ‘Despite the dynamic nature of polish was assumed by use-wear analysts, no models of polish evolution have been proposed, apart from the very general P. Vaughan’s three categories. This lack of models explains two weaknesses of qualitative methods with respect to the identification of worked materials: 1) the capacity to discriminate the worked materials in the first phases of use (generic weak polish) and 2) the overlapping of different types of polishes in the more advanced phases of development (i.e. bone/antler or bone/wood). These two difficulties have been quantitatively demonstrated in our study. For the first time, we have defined specific models of polish development for the five worked materials up to one hour of work. Our quantitative models combining type of worked material and phase of development allow partially overcoming these limitations. First, it is possible discriminate the five worked materials in the first phases of use, at least after 10 minutes of work. Second, in our models, overlapping of worked materials is restricted to antler/bone between 40 and 50 minutes of work. 

I am also still not convinced that we can speak of a "correct identification" of polishes: yes, of course in experimental context we can, but our inferences on archaeological tool use are based on analogies. Past technologies were most likely far more complex than we can imagine and our experiments are often way too simplistic: there are several archaeological polish types for which we have yet to find an experimental equivalent, even after decades of experimental exploration.

After the first analysis the "success rate" is presented to be around 60%, if I understand the text well. The statement that after a second round of statistical analysis virtually all classifications were correct, does not entirely convince me as this is based on a combined analysis of length of use with contact material. Length of use is, as the authors themselves acknowledge, not the only factor behind intensity of polishes (strength, skill, fatigue etc).

A paragraph has been added: ‘We have shown that the analysis of polish texture allows the identification of the intensity of use. This opens a new field of research that should be considered with care. We have used the time of use as the vector defining the intensity of work. However, at the same time, we have shown that hafted an unhafted tools used for the same lapse of time bear different intensities of polish development. Other factors besides hafting, as the strength or experience of the worker, the way in which the worked material is hold (i.e. a hide in a frame or on a surface) and so on can affect the intensity of use. This question opens a new challenge for quantitative use-wear analysis. If the time of work is not the only factor affecting the intensity of use (and the texture modification of polish), how can we build precise dynamic models of polish development? For example, a specific degree of textural modification can correspond to working during 50 minutes with an unhafted tool or for15 minutes with a hafted one. We have defined the intensity of use as steps of 10 minutes of work. Most probably, future dynamic models of use-wear polish will be defined as multifactorial regressions of textural parameters, so tools of unknown intensity of use would be identified with a relative point of intensity of work in the regression model. In anyway, polish texture should be used to compare different intensities of use only IF the rest of the variables characterizing work are similar (i.e. sickles used for harvesting).’ 

In conclusion my advise is "accept with minor revision". Although not questioning the usefulness of the quantitative approach presented in this paper, I think the paper would benefit from a short critical appraisal of its potential and limitations in the conclusion of the paper.

Potential and limitations have been now made apparent in the conclusion of the paper: ‘Some of the aspects which need further research are shared with qualitative use-wear analysis, as the need for more realistic experiments, reproducing the enormous variability of working procedures in the Prehistoric past.’ […] ‘Other issues that should be developed are specific for the quantitative studies. In order to standardize the analysis, it is necessary to reach a consensus between specialists on the technical aspects of the methodology (microscope proprieties and chosen techniques of analysis, magnifications, filtering procedure, parameters of texture and so on). Thus, we are convinced that the further development of quantitative methods imply the collaboration of several research groups.’

---

## [Decision Letter · Decision Letter 1]

28 Jul 2021

PONE-D-21-14694R1

Quantitative Use-wear Analysis of Stone Tools: Measuring How the Intensity of Use Affects the Identification of the Worked Material

PLOS ONE

Dear Dr. Ibanez,

Thank you for submitting your manuscript to PLOS ONE. After careful consideration, we feel that it has merit but does not fully meet PLOS ONE’s publication criteria as it currently stands. Therefore, we invite you to submit a revised version of the manuscript that addresses the points raised during the review process. In particular, please check the complete reference list, as suggested by Reviewer 2. 

We look forward to receiving your revised manuscript.

Kind regards,

Enza Elena Spinapolice, Ph.D

Academic Editor

PLOS ONE

Journal Requirements:

Reviewers' comments:

Reviewer's Responses to Questions

**Comments to the Author**

1. If the authors have adequately addressed your comments raised in a previous round of review and you feel that this manuscript is now acceptable for publication, you may indicate that here to bypass the “Comments to the Author” section, enter your conflict of interest statement in the “Confidential to Editor” section, and submit your "Accept" recommendation.

Reviewer #1: All comments have been addressed

Reviewer #2: All comments have been addressed

2. Is the manuscript technically sound, and do the data support the conclusions?

Reviewer #1: Yes

Reviewer #2: Yes

3. Has the statistical analysis been performed appropriately and rigorously? 

Reviewer #1: Yes

Reviewer #2: I Don't Know

4. Have the authors made all data underlying the findings in their manuscript fully available?

Reviewer #1: (No Response)

Reviewer #2: Yes

5. Is the manuscript presented in an intelligible fashion and written in standard English?

Reviewer #1: Yes

Reviewer #2: Yes

6. Review Comments to the Author

Reviewer #1: (No Response)

Reviewer #2: I did not check all references but in two that I know well I still found mistakes (authors missing, year wrong). Below the correct references:

Guzzo Falci., Cuisin J., Delpuech A., Gijn. A.L. van & Hofman C.L. (2018), New insights into use-wear development in bodily ornaments through the study of ethnographic collections, Journal of Archaeological Method and Theory: 1-51

https://doi.org/10.1007/s10816-018-9389-8

Unrath G., Owen L.R., Gijn A.L. van, Moss E.H., Plisson H. & Vaughan P. (1986), An evaluation of micro-wear studies: a multi-analyst approach.. In: Unrath G., Owen L.R. (Eds.) Technical aspects of micro-wear studies on stone tools.. Tubingen: Archeologica Venatoria. 117-176.

7. PLOS authors have the option to publish the peer review history of their article (what does this mean?). If published, this will include your full peer review and any attached files.

Reviewer #1: No

Reviewer #2: No

---

## [Author Response · Author response to Decision Letter 1]

31 Jul 2021

We have corrected the references as suggested by reviewer 2

---

## [Editor Report · Decision Letter 2]

31 Aug 2021

Quantitative Use-wear Analysis of Stone Tools: Measuring How the Intensity of Use Affects the Identification of the Worked Material

PONE-D-21-14694R2

Dear Dr. Ibanez,

We’re pleased to inform you that your manuscript has been judged scientifically suitable for publication and will be formally accepted for publication once it meets all outstanding technical requirements.

Kind regards,

Enza Elena Spinapolice, Ph.D

Academic Editor

PLOS ONE

Additional Editor Comments (optional):

Please check and edit the reference list.

Here some suggestions from the reviewer 2

I did not check all references but in two that I know well I still found mistakes (authors missing, year wrong). Below the correct references:

Guzzo Falci., Cuisin J., Delpuech A., Gijn. A.L. van & Hofman C.L. (2018), New insights into use-wear development in bodily ornaments through the study of ethnographic collections, Journal of Archaeological Method and Theory: 1-51

https://doi.org/10.1007/s10816-018-9389-8

Unrath G., Owen L.R., Gijn A.L. van, Moss E.H., Plisson H. & Vaughan P. (1986), An evaluation of micro-wear studies: a multi-analyst approach.. In: Unrath G., Owen L.R. (Eds.) Technical aspects of micro-wear studies on stone tools.. Tubingen: Archeologica Venatoria. 117-176.
---

## [Editor Report · Acceptance letter]

2 Sep 2021

PONE-D-21-14694R2 

Quantitative Use-wear Analysis of Stone Tools: Measuring How the Intensity of Use Affects the Identification of the Worked Material 

Dear Dr. Ibáñez:

I'm pleased to inform you that your manuscript has been deemed suitable for publication in PLOS ONE. Congratulations! Your manuscript is now with our production department. 

Kind regards, 

on behalf of

Dr. Enza Elena Spinapolice 

Academic Editor

PLOS ONE